# Relational Self-Attention:
# What's Missing in Attention for Video Understanding

**Manjin Kim**[1][*]   **Heeseung Kwon**[1][*]   **Chunyu Wang**[2]   **Suha Kwak**[1]   **Minsu Cho**[1]

[1]POSTECH        [2]Microsoft Research Asia

http://cvlab.postech.ac.kr/research/RSA/

## Abstract

Convolution has been arguably the most important feature transform for modern neural networks, leading to the advance of deep learning. Recent emergence of Transformer networks, which replace convolution layers with self-attention blocks, has revealed the limitation of stationary convolution kernels and opened the door to the era of dynamic feature transforms. The existing dynamic transforms, including self-attention, however, are all limited for video understanding where correspondence relations in space and time, *i.e.*, motion information, are crucial for effective representation. In this work, we introduce a relational feature transform, dubbed the *relational self-attention (RSA)*, that leverages rich structures of spatio-temporal relations in videos by dynamically generating relational kernels and aggregating relational contexts. Our experiments and ablation studies show that the RSA network substantially outperforms convolution and self-attention counterparts, achieving the state of the art on the standard motion-centric benchmarks for video action recognition, such as Something-Something-V1&V2, Diving48, and FineGym.

## 1   Introduction

Convolution [13, 25] is a feature transform that is ubiquitous in modern neural networks for visual recognition, and has driven the development of deep learning in the past decade. The stationarity of convolution kernels, however, may limit its expressivity and hinder adaptation to diverse compositional possibilities of visual concepts [18]. Dynamic convolution [20] and self-attention [52] that construct kernels or attentions according to input contents have emerged as an alternative to static convolution in this context, being followed by further studies for dynamic feature transforms [6, 27, 33, 57]. The effectiveness of self-attention has been demonstrated by the success of Transformer variants on different image understanding tasks such as image classification [7, 51, 59], object detection [4], and semantic segmentation [43]. Recently, it has been further extended for video understanding, replacing spatio-temporal convolution [5, 48, 50] with spatio-temporal self-attention [1, 3, 35].

Despite their recent progress in the video domain, the existing dynamic feature transforms still leave much room for improvement in terms of learning relational patterns in space and time, *i.e.*, motion information, which is known to be essential for video understanding [23, 53]. For example, spatio-temporal self-attention [55] often fails to learn motion representation without positional embeddings as demonstrated in [32], and even those with positional embeddings [1, 3, 35] turn out to be not effective on motion-centric action recognition benchmarks such as Something-Something [14].

In this work, we introduce a relational dynamic feature transform, dubbed *relational self-attention (RSA)*, to address the limitation of existing methods. RSA leverages rich structures of spatio-temporal

---

[*]Equal contribution.

35th Conference on Neural Information Processing Systems (NeurIPS 2021).

relations in videos by dynamically generating relational kernels and aggregating relational contexts. Combining these relational components with ordinary dynamic kernels and context features, our RSA learns a rich video representation that effectively captures both visual appearance and spatio-temporal motion dynamics.

The main contribution of this paper is three-fold:

- We re-interpret recent dynamic feature transforms in a unified way, and provide in-depth analysis on their capability of learning video representation.

- We introduce a new dynamic feature transform, *i.e.*, RSA, which effectively captures both visual appearance and spatio-temporal motion dynamics for video understanding.

- Our RSA network substantially outperforms convolution and self-attention counterparts, achieving the state of the art on SS-V1&V2, Diving48, and FineGym.

## 2  Related Work

**Convolution and its variants.** Convolution [13, 25] has been used as a dominant neural primitive in modern neural architectures [17, 34, 38, 42, 45, 47]. Since convolution often suffers from its limited expressivity due to its static kernel, dynamic convolution operators have been studied for enhancing composability by dynamically adjusting the convolution kernel according to input features [6, 20, 27, 33, 57]. One example is involution [27], which generates a lightweight dynamic kernel using input content and substantially outperforms convolution with less computational cost on image classification. Our RSA differs from these kernel generation methods in that it leverages relational patterns for learning a rich video representation.

**Self-attention and its variants.** Self-attention [52] was originally introduced for neural machine translation to capture long-range interactions, and now has been widely adopted in many different domains thanks to its versatility and expandability. Following local attention methods [18, 37] that employ self-attention as an alternative to convolution, ViT models have demonstrated impressive performance on a variety of image understanding tasks [4, 7, 43, 59]. On the other hand, some of previous work have attempted to design novel dynamic transforms through attention [2, 60]. Zhao *et al.* [60] expand the spatial attention to include the channel-wise attention. Bello [2] proposes a lambda layer that focuses on interaction between visual contents and a relative position embedding without softmax, which outperforms self-attention counterparts on image classification. The proposed RSA is an extension of these techniques, yet focuses on learning rich relational features for video understanding.

**Convolution and self-attention for video understanding.** Action recognition is the most fundamental task for video understanding, which aims at classifying a video clip into pre-defined action classes. A key to success of this task is to capture temporal dynamics across multiple video frames, and spatio-temporal (3D) convolution [48] has been a *de facto* for modeling the temporal dynamics [8, 10, 31, 49, 50]. On the other hand, self-attention was used as an attachable module for 3D CNNs in the earliest method [55], yet nowadays becomes the major building block of video understanding networks such as video ViT [1, 3, 35]. However, both 3D CNNs and ViTs are not sufficient for modeling temporal dynamics due to the lack of inter-feature relations within the spatio-temporal context. Our RSA overcomes their limitation by explicitly utilizing relational features that imply temporal dynamics.

**Learning motion for video understanding.** Early approaches in video understanding rely on external motion extractors like optical flow for motion feature learning [5, 12, 41], but recent models aim to learn motion representation internally. To this end, they estimate feature-level optical flow inside their networks [9, 36] or compute subtraction between consecutive frame-wise features [21, 26, 29, 44]. Meanwhile, correlation-based methods [23, 24, 53] use inter-pixel relations as motion likelihood maps and achieve state-of-the-art on motion-centric action recognition benchmarks. Inspired by the correlation-based methods, we apply dynamic relational kernels to the correlation tensors (*i.e.*, relational context), leading to effective motion feature learning.

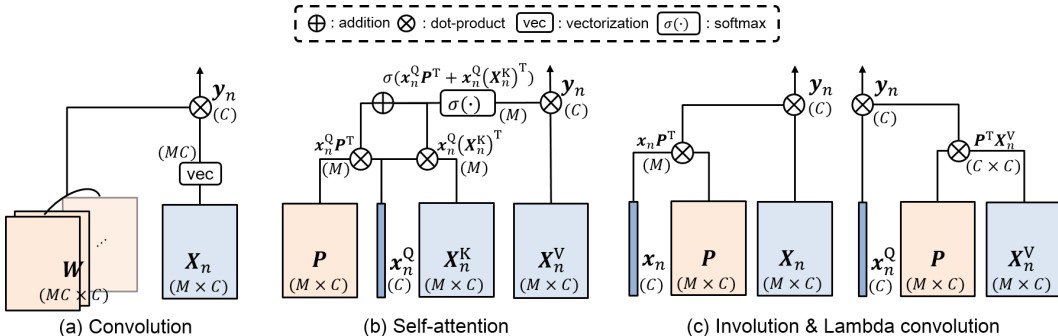

Figure 1: **Computational graphs of different feature transforms.** See text for details.

## 3 Background

In this section, we revisit three instances of feature transform and analyze them in a simplified form. In our context, the role of feature transform is to update a feature map while preserving its order in space and/or time. To this end, it processes each position in space and/or time using features of its neighborhood. For example, given an input feature map of a video $\boldsymbol{X} \in \mathbb{R}^{T \times H \times W \times C}$, for each spatio-temporal position $(t, h, w)$, a feature transform takes the individual feature $\boldsymbol{x}_{t,h,w}$ as a *target* and transforms it using its neighborhood features $\{\boldsymbol{x}_{t',h',w'}\}_{(t',h',w') \in \mathcal{N}_{(t,h,w)}}$ as its *context*.

For the sake of notational convenience, let us denote the target by $\boldsymbol{x}_n \in \mathbb{R}^C$, where $n$ represents a specific position in space and/or time, and its context by $\boldsymbol{X}_n \in \mathbb{R}^{M \times C}$, where $M$ is the size of neighborhood, and the corresponding output by $\boldsymbol{y}_n \in \mathbb{R}^C$. Now, without loss of generality, we can view the *transform* as applying to each position $n$ a function $f$ with learnable weights $\boldsymbol{W}$ that maps target $\boldsymbol{x}_n$ to output $\boldsymbol{y}_n$ using context $\boldsymbol{X}_n$: $\boldsymbol{y}_n = f(\boldsymbol{x}_n, \boldsymbol{X}_n; \boldsymbol{W})$. Note that such a transform is typically implemented as a neural layer that processes all the targets in parallel.

In the following, we describe recent feature transform functions in computer vision as well as the conventional one, convolution. Figure 1 illustrates the computational graphs of the transforms.

**Convolution.** Convolution [13, 25] is a static and translation-equivariant feature transform that updates each target $\boldsymbol{x}_n$ by applying static kernels $\boldsymbol{W} \in \mathbb{R}^{MC \times C}$ on local context $\boldsymbol{X}_n$:

$$\boldsymbol{y}_n = \boldsymbol{W}^\mathsf{T} \text{vec}(\boldsymbol{X}_n). \tag{1}$$

While extracting different visual patterns using multiple kernels, convolution remains static in the sense that the kernel is not affected by the target $\boldsymbol{x}_n$, *i.e.*, feature of interest (Figure 1a). This stationarity may not be effective in adapting to diverse compositional possibilities of visual concepts [18], and the channel-dependent weights can cause inter-channel redundancy in the kernel [19].

**Self-attention.** Self-attention [52] is a dynamic transform that generates an attention map of context $\boldsymbol{X}_n$ using target $\boldsymbol{x}_n$ and then aggregates the context using the attention map as a dynamic kernel (Figure 1b). The process starts by adapting inputs for query-key-value attention computation; using three learnable embedding matrices, $\boldsymbol{E}^\mathrm{Q}, \boldsymbol{E}^\mathrm{K}, \boldsymbol{E}^\mathrm{V} \in \mathbb{R}^{C \times C}$, it embeds the target $\boldsymbol{x}_n$ into the query $\boldsymbol{x}_n^\mathrm{Q}$ via $\boldsymbol{x}_n^\mathrm{Q} = \boldsymbol{x}_n \boldsymbol{E}^\mathrm{Q}$ and also projects the context $\boldsymbol{X}_n$ into the key $\boldsymbol{X}_n^\mathrm{K}$ and the value $\boldsymbol{X}_n^\mathrm{V}$ via $\boldsymbol{X}_n^\mathrm{K} = \boldsymbol{X}_n \boldsymbol{E}^\mathrm{K}$ and $\boldsymbol{X}_n^\mathrm{V} = \boldsymbol{X}_n \boldsymbol{E}^\mathrm{V}$, respectively. The basic attention map is computed via the softmaxed product between query $\boldsymbol{x}_n^\mathrm{Q}$ and key $\boldsymbol{X}_n^\mathrm{K}$, which can be viewed as *content-to-content interaction*. To reinforce this content-only attention with positional information of the context, a learnable matrix $\boldsymbol{P} \in \mathbb{R}^{M \times C}$, which encodes relative positions of individual features in the context, is optionally included as *content-to-position interaction*. The self-attention transform aggregates the value embeddings $\boldsymbol{X}_n^\mathrm{V}$ of context using the attention map as kernel weights shared for all channels:

$$\boldsymbol{y}_n = \sigma(\boldsymbol{x}_n^\mathrm{Q}(\boldsymbol{X}_n^\mathrm{K})^\mathsf{T} + \boldsymbol{x}_n^\mathrm{Q} \boldsymbol{P}^\mathsf{T}) \boldsymbol{X}_n^\mathrm{V}, \tag{2}$$

where $\sigma(\cdot)$ denote the softmax function. This transform is translation-equivariant if position embedding $\boldsymbol{P}$ is local and relative with respect to the target, and becomes permutation-equivariant if content-to-position interaction is removed. Unlike convolution, it is *dynamic* in the sense that the kernel $\sigma(\boldsymbol{x}_n^\mathrm{Q}(\boldsymbol{X}_n^\mathrm{K})^\mathsf{T} + \boldsymbol{x}_n^\mathrm{Q} \boldsymbol{P}^\mathsf{T})$, which is used to aggregate the context, depends on the target content.

It allows more flexibility in adapting to input and also consumes fewer parameters by sharing the kernel across the channels [18].

**Involution & lambda convolution.** Involution [27] is a light-weight dynamic transform that leverages *content-to-position interaction* only. It dynamically generates a kernel $\boldsymbol{\kappa}_n^{\mathrm{V}} \in \mathbb{R}^M$ by projecting target $\boldsymbol{x}_n$ using a learnable matrix $\boldsymbol{P} \in \mathbb{R}^{M \times C}$ via $\boldsymbol{\kappa}_n^{\mathrm{V}} = \boldsymbol{x}_n \boldsymbol{P}^{\mathsf{T}}$, and then use the kernel to aggregate the context $\boldsymbol{X}_n$ (Figure 1c, left):

$$\boldsymbol{y}_n = \boldsymbol{\kappa}_n^{\mathrm{V}} \boldsymbol{X}_n = \boldsymbol{x}_n \boldsymbol{P}^{\mathsf{T}} \boldsymbol{X}_n, \tag{3}$$

where the matrix $\boldsymbol{P}$ plays the role of converting the target to a kernel in a position-sensitive manner. In a similar spirit, lambda convolution[2] of the lambda networks [2] uses *content-to-position interaction* between the target and the context position. As in self-attention, using two learnable embedding matrices, $\boldsymbol{E}^{\mathrm{Q}}, \boldsymbol{E}^{\mathrm{V}} \in \mathbb{R}^{C \times C}$, it starts by projecting the target $\boldsymbol{x}_n$ and the context $\boldsymbol{X}_n$ into the query $\boldsymbol{x}_n^{\mathrm{Q}}$ and the value $\boldsymbol{X}_n^{\mathrm{V}}$ via $\boldsymbol{x}_n^{\mathrm{Q}} = \boldsymbol{x}_n \boldsymbol{E}^{\mathrm{Q}}$ and $\boldsymbol{X}_n^{\mathrm{V}} = \boldsymbol{X}_n \boldsymbol{E}^{\mathrm{V}}$, respectively. The lambda convolution abstracts the context value $\boldsymbol{X}_n^{\mathrm{V}}$ to a contextual matrix $\boldsymbol{\lambda}_n^{\mathrm{p}}$ using a learnable matrix $\boldsymbol{P} \in \mathbb{R}^{M \times C}$ via $\boldsymbol{\lambda}_n^{\mathrm{p}} = \boldsymbol{P}^{\mathsf{T}} \boldsymbol{X}_n^{\mathrm{V}} \in \mathbb{R}^{C \times C}$, which in turn is used for updating the target query $\boldsymbol{x}_n^{\mathrm{Q}}$ (Figure 1c, right):

$$\boldsymbol{y}_n = \boldsymbol{x}_n^{\mathrm{Q}} \boldsymbol{\lambda}_n^{\mathrm{p}} = \boldsymbol{x}_n^{\mathrm{Q}} \boldsymbol{P}^{\mathsf{T}} \boldsymbol{X}_n^{\mathrm{V}}. \tag{4}$$

Despite the differences in the operational procedures and the concepts, the lambda convolution has effectively the same form with involution except for additional key and value embeddings. Note that unlike self-attention, involution and lambda convolution both have no softmax nonlinearity. In fact, the absence of softmax reduces the computational cost and also increases the expressive ability by allowing negative activations [27]. Both involution and lambda convolution are shown to outperform convolution and self-attention counterparts in image classification [2, 27], demonstrating the effectiveness of dynamic content-aware kernels.

**Limitation of existing dynamic transforms.** The aforementioned dynamic transforms are commonly based on leveraging content-to-content and/or content-to-position interactions in constructing kernels, where the target content (input feature) is used as the source of the dynamic transform. While the methods are effective for learning image representation indeed, they are all limited for learning video representation; as will be shown in the experimental section 5.3, we have found that existing dynamic transforms show only marginal or virtually no improvement over the static transform of convolution on the motion-centric action recognition benchmark. The main reason lies in the missing *structure* of content-to-content interactions as a *relational content* in representation learning. While content-to-content interactions are considered in existing dynamic transforms, they are immediately broken into individuals without being used as a whole. For example, self-attention computes query-to-key correlation $\boldsymbol{x}_n^{\mathrm{Q}} (\boldsymbol{X}_n^{\mathrm{K}})^{\mathsf{T}}$ but uses the individual elements of the correlation only for aggregating informative contents from the context. The content-to-position interaction $\boldsymbol{x}_n^{\mathrm{Q}} \boldsymbol{P}^{\mathsf{T}}$ does not help in capturing the structure of interactions either since it has no access to the correlation as a whole. In videos, such structural patterns contain informative spatio-temporal contents, *i.e.*, different levels of generic motion information, thus being crucial in video understanding.

## 4   Our approach

In this section, we introduce a new dynamic transform, dubbed *relational self-attention (RSA)*, which is designed to learn rich spatio-temporal interaction patterns across input contents. Figure 2 illustrates the computational graph of RSA. On top of the basic kernel and context in a dynamic transform, it builds relational kernel and context and processes all the kernel-context combinations. Here we describe the main components of RSA and their integrated form, and explain an efficient implementation of RSA.

As in self-attention, we start by adapting inputs for query-key-value interactions; using three learnable embedding matrices, $\boldsymbol{E}^{\mathrm{Q}}, \boldsymbol{E}^{\mathrm{K}}, \boldsymbol{E}^{\mathrm{V}} \in \mathbb{R}^{C \times C}$, target $\boldsymbol{x}_n$ and context $\boldsymbol{X}_n$ are embedded into query $\boldsymbol{x}_n^{\mathrm{Q}}$, key $\boldsymbol{X}_n^{\mathrm{K}}$ and value $\boldsymbol{X}_n^{\mathrm{V}}$ via $\boldsymbol{x}_n^{\mathrm{Q}} = \boldsymbol{x}_n \boldsymbol{E}^{\mathrm{Q}}$, $\boldsymbol{X}_n^{\mathrm{K}} = \boldsymbol{X}_n \boldsymbol{E}^{\mathrm{K}}$, and $\boldsymbol{X}_n^{\mathrm{V}} = \boldsymbol{X}_n \boldsymbol{E}^{\mathrm{V}}$, respectively. $\boldsymbol{x}_n^{\mathrm{Q}}, \boldsymbol{X}_n^{\mathrm{K}}$, and $\boldsymbol{X}_n^{\mathrm{V}}$ are then L2-normalized, and we omit the normalization term for the simplicity.

---

[2]This transform corresponds to the position lambda where the extent of the context $\boldsymbol{X}_n^{\mathrm{V}}$ is restricted to the local neighborhood of the target $\boldsymbol{x}_n^{\mathrm{Q}}$ [2].

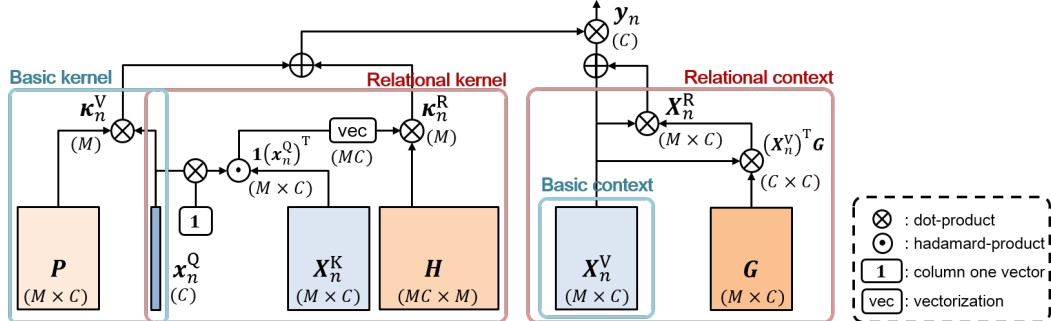

Figure 2: **Computational graph of RSA.** RSA consists of two types of kernels (basic and relational kernel) and two contexts (basic and relational context). See text for details.

## 4.1 Relational self-attention (RSA)

**Relational kernel.** The relational kernel is designed to predict the relevance of context based on the structure of content-to-content interactions. To generate the relational kernel, we compute the dot-product correlation of the query and the key, *i.e.*, $x_n^Q(X_n^K)^\mathsf{T} \in \mathbb{R}^M$, and then project the correlation vector using a learnable matrix $H \in \mathbb{R}^{M \times M}$:

$$\kappa_n^R = x_n^Q(X_n^K)^\mathsf{T} H. \tag{5}$$

The role of $H$ in this relational kernel corresponds to that of $P$ in the involution kernel (Eq. 3); while $P$ predicts the kernel weights from the $C$-dimensional query vector, $H$ predicts them from the $M$-dimensional query-key correlation vector. The resultant dynamic kernel aggregates the context depending on how the individual contents are related to each other in space and time, thus being particularly effective in learning motion-related patterns in videos. Note that, when we set $H$ to an identity matrix $I$ and add the softmax operator into Eq. 5, the relational kernel $\kappa_n^R$ is equivalent to the self-attention kernel without content-to-position interaction, *i.e.*, $\sigma(x_n^Q(X_n^K)^\mathsf{T})$.

Since the dot-product correlation contracts all channel dimensions of the query and the keys, we may lose semantic information, which may help in generating an effective relational kernel. We thus take the Hadamard product [22] instead so that we can leverage channel-wise query-key correlations for producing the relational kernel. Using a learnable kernel projection matrix $H \in \mathbb{R}^{MC \times M}$, Eq. 5 can be reformulated as

$$\kappa_n^R = \text{vec}(1(x_n^Q)^\mathsf{T} \odot X_n^K)H, \tag{6}$$

where $\odot$ denotes the Hadamard product. $H \in \mathbb{R}^{MC \times M}$ predicts the kernel weights from an $MC$-dimensional channel-wise query-key correlation vector. Furthermore, the absence of the softmax operator allows the relational kernel to have negative activations so that it better learns relative features where the subtractive interactions are beneficial for capturing relative changes of contents over time [21, 26, 29].

**Relational context.** The relational context is designed to provide the relational pattern of content-to-content (*i.e.*, context-to-context) interactions for the kernels to aggregate. To this end, we use self-correlation [40], which effectively describes spatio-temporal intra-structure including motion information [23, 24, 53]. We first construct the self-correlation matrix $X_n^V(X_n^V)^\mathsf{T} \in \mathbb{R}^{M \times M}$ and then project it using a learnable matrix $G \in \mathbb{R}^{M \times C}$ into the relational context $X_n^R$:

$$X_n^R = X_n^V(X_n^V)^\mathsf{T} G, \tag{7}$$

where the self-correlation $X_n^V(X_n^V)^\mathsf{T}$ reveals content-to-content interaction patterns within the context, and the matrix $G$ maps it to the relational context so that the output has the same size with the basic context $X_n^V$.

**Combining of different types of kernels and contexts.** The proposed transform, *RSA*, integrates the relational kernel and context into a dynamic transform. As illustrated in Figure 2, it consists of two types of kernels, $\kappa_n^V$ and $\kappa_n^R$, and two types of contexts, $X_n^V$ and $X_n^R$. Note that the basic kernel $\kappa_n^V$ is computed as $x_n^Q P^\mathsf{T}$. We combine the kernels and the contexts in the RSA transform:

$$\begin{aligned} y_n &= (\kappa_n^V + \kappa_n^R)(X_n^V + X_n^R) \\ &= \kappa_n^V X_n^V + \kappa_n^R X_n^V + \kappa_n^V X_n^R + \kappa_n^R X_n^R, \end{aligned} \tag{8}$$

Table 1: **Complexity of RSA.** $B, N, M, C, D, L$ denotes batch size, input size, context size, channel dimension, latent channel dimension, and number of queries, respectively. We simplify the complexity terms using $M \gg L, D \leq C/L$ for ease description.

| operation | time complexity | space complexity |
|---|---|---|
| RSA | $\mathcal{O}(BNM^2C)$ | $\mathcal{O}(BNM(M+C) + M^2C)$ |
| + Efficient RSA | $\mathcal{O}(BNMC^2)$ | $\mathcal{O}(BNC^2 + MC^2)$ |
| + Multi-query ($L$) | $\mathcal{O}(BNMC^2/L^2)$ | $\mathcal{O}(BNC^2/L^2 + BNC + MC^2/L^2)$ |

which contains four different dynamic transforms, $\boldsymbol{\kappa}_n^{\mathrm{V}}\boldsymbol{X}_n^{\mathrm{V}}$, $\boldsymbol{\kappa}_n^{\mathrm{R}}\boldsymbol{X}_n^{\mathrm{V}}$, $\boldsymbol{\kappa}_n^{\mathrm{V}}\boldsymbol{X}_n^{\mathrm{R}}$, and $\boldsymbol{\kappa}_n^{\mathrm{R}}\boldsymbol{X}_n^{\mathrm{R}}$. The first sub-transform, $\boldsymbol{\kappa}_n^{\mathrm{V}}\boldsymbol{X}_n^{\mathrm{V}}$ corresponds to the involution (Eq. 3). The second and third sub-transform, $\boldsymbol{\kappa}_n^{\mathrm{R}}\boldsymbol{X}_n^{\mathrm{V}}$ and $\boldsymbol{\kappa}_n^{\mathrm{V}}\boldsymbol{X}_n^{\mathrm{R}}$, both capture content-to-relation interactions, but their effects are different; while $\boldsymbol{\kappa}_n^{\mathrm{R}}\boldsymbol{X}_n^{\mathrm{V}}$ aggregates the basic context with considering the content-to-content interactions, $\boldsymbol{\kappa}_n^{\mathrm{V}}\boldsymbol{X}_n^{\mathrm{R}}$ aggregates the relational patterns with the content-to-position interactions. The last sub-transform, $\boldsymbol{\kappa}_n^{\mathrm{R}}\boldsymbol{X}_n^{\mathrm{R}}$, is fully dependent on the relational information. The relational kernel dynamically aggregates relational patterns, generating the deeper relational interactions. These sub-transforms altogether encode rich relational interactions, leading to comprehensive video representation learning.

## 4.2 Improving efficiency of RSA

**Efficient relational kernel.** The projection matrix $\boldsymbol{H} \in \mathbb{R}^{MC \times M}$ of the relational kernel increases the cost of computation and memory quadratically with the context size $M$, causing a computational bottleneck. We reduce the complexity to the linear one by decomposing $\boldsymbol{H}$ to $\boldsymbol{H}_1\boldsymbol{H}_2^{\mathsf{T}}$ such that $\boldsymbol{H}_1 \in \mathbb{R}^{MC \times D}$ and $\boldsymbol{H}_2 \in \mathbb{R}^{M \times D}$, where $D$ is a latent channel dimension, which is smaller than $M$. Furthermore, we dramatically reduce the memory footprint by switching the computation orders. When $r(\boldsymbol{H}_1) \in \mathbb{R}^{M \times C \times D}$ is the reshaped $\boldsymbol{H}_1 \in \mathbb{R}^{MC \times D}$, the kernel equation is re-formulated as

$$\boldsymbol{\kappa}_n^{\mathrm{R}} = \mathrm{vec}(\mathbf{1}(\boldsymbol{x}_n^{\mathrm{Q}})^{\mathsf{T}} \odot \boldsymbol{X}_n^{\mathrm{K}})\boldsymbol{H}_1\boldsymbol{H}_2^{\mathsf{T}} \tag{9}$$

$$= \boldsymbol{x}_n^{\mathrm{Q}}(\boldsymbol{X}_n^{\mathrm{K}} \circledast r(\boldsymbol{H}_1))\boldsymbol{H}_2^{\mathsf{T}}, \quad \text{where } (\boldsymbol{X}_n^{\mathrm{K}} \circledast r(\boldsymbol{H}_1))_{c,d} = \sum_m (\boldsymbol{X}_n^{\mathrm{K}})_{m,c}(r(\boldsymbol{H}_1))_{m,c,d}. \tag{10}$$

Note that $\boldsymbol{X}_n^{\mathrm{K}} \circledast r(\boldsymbol{H}_1)$ can be efficiently implemented by a channel-wise convolution. Our experiments show that this modification achieves a good computation-complexity trade-off (Table 4b). Please refer to the pseudo code Fig. 1 in our Supp. for more details.

**Efficient RSA.** The vanilla RSA may require a high cost of computation and memory when generating contexts and aggregating the kernels with the contexts. To reduce them, we decompose $\boldsymbol{P}$ to $\boldsymbol{H}_2\boldsymbol{P}_1$ such that $\boldsymbol{H}_2 \in \mathbb{R}^{M \times D}, \boldsymbol{P}_1 \in \mathbb{R}^{D \times C}$, where $\boldsymbol{H}_2$ is the same one obtained by decomposing $\boldsymbol{H}$. The RSA equation is then re-formulated as

$$\begin{aligned}
\boldsymbol{y}_n &= (\boldsymbol{\kappa}_n^{\mathrm{V}} + \boldsymbol{\kappa}_n^{\mathrm{R}})(\boldsymbol{X}_n^{\mathrm{V}} + \boldsymbol{X}_n^{\mathrm{R}}) \\
&= (\boldsymbol{x}_n^{\mathrm{Q}}\boldsymbol{P}^{\mathsf{T}} + \boldsymbol{x}_n^{\mathrm{Q}}(\boldsymbol{X}_n^{\mathrm{K}} \circledast r(\boldsymbol{H}_1))\boldsymbol{H}_2^{\mathsf{T}})(\boldsymbol{X}_n^{\mathrm{V}} + \boldsymbol{X}_n^{\mathrm{V}}(\boldsymbol{X}_n^{\mathrm{V}})^{\mathsf{T}}\boldsymbol{G}) \tag{11} \\
&= \boldsymbol{x}_n^{\mathrm{Q}}(\boldsymbol{P}^{\mathsf{T}} + (\boldsymbol{X}_n^{\mathrm{K}} \circledast r(\boldsymbol{H}_1))\boldsymbol{H}_2^{\mathsf{T}})\boldsymbol{X}_n^{\mathrm{V}}(\boldsymbol{I} + (\boldsymbol{X}_n^{\mathrm{V}})^{\mathsf{T}}\boldsymbol{G}) \\
&= \boldsymbol{x}_n^{\mathrm{Q}}(\boldsymbol{P}_1^{\mathsf{T}} + \boldsymbol{X}_n^{\mathrm{K}} \circledast r(\boldsymbol{H}_1))(\boldsymbol{H}_2^{\mathsf{T}}\boldsymbol{X}_n^{\mathrm{V}})(\boldsymbol{I} + (\boldsymbol{X}_n^{\mathrm{V}})^{\mathsf{T}}\boldsymbol{G}) \quad \text{where } \boldsymbol{P} = \boldsymbol{H}_2\boldsymbol{P}_1. \tag{12}
\end{aligned}$$

Note that $\boldsymbol{X}_n^{\mathrm{K}} \circledast r(\boldsymbol{H}_1)$, $\boldsymbol{H}_2^{\mathsf{T}}\boldsymbol{X}_n^{\mathrm{V}}$, and $(\boldsymbol{X}_n^{\mathrm{V}})^{\mathsf{T}}\boldsymbol{G}$ can be efficiently implemented with convolutions. As shown in Table 1, the space and time complexities are respectively reduced to $\mathcal{O}(BNMC^2)$ and $\mathcal{O}(BNC^2 + MC^2)$, which are both linear to $M$. Note that the space complexity is proportional to $N$ and $M$ separately, so that it facilitates efficient training with sufficiently a large volume of context size $M$, $e.g.$, $5 \times 7 \times 7$. Please refer to the pseudo code Fig. 2 in our Supp. for more details.

**Multi-query RSA.** We adopt the multi-query setting [2] for RSA, which sets $L$ number of multiple queries and applies the same key and value context to each query, where the size of channel dimension of each query $C^{\mathrm{Q}}$ becomes $C/L$. While the multi-head setting [52] maintains the time complexity and increases the space complexity, multi-query setting significantly reduces both of them (Table 1).

Table 2: **Performance comparison with other spatio-temporal feature transform methods on SS-v1**. $\boldsymbol{W}_j \circ \boldsymbol{W}_i(\cdot)$ indicates a sequential transform of $\boldsymbol{W}_i$ followed by $\boldsymbol{W}_j$. $\sigma$ denotes softmax.

| feature transform | kernel | context | FLOPs | params. | top-1 | top-5 |
|---|---|---|---|---|---|---|
| 2D convolution | $\boldsymbol{W}_{2D,standard}$ | $\boldsymbol{X}_n$ | 32.5 G | 24.3 M | 19.7 | 46.6 |
| 3D convolution [48] | $\boldsymbol{W}_{3D,standard}$ | $\boldsymbol{X}_n$ | 57.0 G | 39.3 M | 43.3 | 72.2 |
| (2+1)D convolution [50] | $\boldsymbol{W}_{1D,standard} \circ \boldsymbol{W}_{2D,standard}$ | $\boldsymbol{X}_n$ | 37.3 G | 26.8 M | 44.1 | 72.9 |
| Self-attention [37] | $\sigma(\boldsymbol{x}_n^Q(\boldsymbol{X}_n^K)^\mathsf{T} + \boldsymbol{x}_n^Q \boldsymbol{P}^\mathsf{T}))$ | $\boldsymbol{X}_n^V$ | 32.2 G | 23.4 M | 41.6 | 70.9 |
| Self-attention variant [37] | $\sigma(\boldsymbol{x}_n^Q(\boldsymbol{X}_n^K)^\mathsf{T})$ | $\boldsymbol{X}_n^V$ | 32.1 G | 23.4 M | 25.9 | 56.3 |
| Self-attention variant [37] | $\sigma(\boldsymbol{x}_n^Q \boldsymbol{P}^\mathsf{T})$ | $\boldsymbol{X}_n^V$ | 32.1 G | 23.4 M | 41.3 | 70.6 |
| Self-attention variant [37] | $\boldsymbol{x}_n^Q \boldsymbol{P}^\mathsf{T}$ | $\boldsymbol{X}_n^V$ | 32.1 G | 23.4 M | 44.3 | 73.7 |
| RSA (ours) | $\boldsymbol{\kappa}_n^R + \boldsymbol{\kappa}_n^V$ | $\boldsymbol{X}_n^V$ | 32.6 G | 23.6 M | 45.7 | 74.8 |
| RSA (ours) | $\boldsymbol{\kappa}_n^R + \boldsymbol{\kappa}_n^V$ | $\boldsymbol{X}_n^V + \boldsymbol{X}_n^R$ | 33.2 G | 23.6 M | **47.0** | **75.7** |

# 5 Experiments

## 5.1 Implementation details

**Architecture details.** We use TSN-ResNet50 [54] as our backbone and replace the standard spatial convolution layers by spatio-temporal RSA layers for every two ResNet bottlenecks [17]. Unless specified otherwise, we replace 7 RSA layers, there are 7 RSA layers in total where $L = 8$, $D = C^Q$, $M$=5×7×7. We set the input and output channel dimensions of RSA layers to be equal to those of spatial convolution layers in TSN-ResNet50.

**Training & testing details.** For initialization, we randomly initialize the weights of bottlenecks including RSA layers with the MSRA method [16] and use ImageNet pre-trained weights for all the other layers. We set the gamma parameter of the last batch normalization layer to zero. For training, we sample 8 or 16 frames from each video using the segment-based sampling strategy [54]. For testing, we sample one or two clips consisting of 8 or 16 frames using the segment-based sampling, and average softmax scores for final prediction. Refer to Sec.1 in our Supp. for more details.

## 5.2 Datasets

**Something-something v1 & v2 (SS-V1 & V2)** [14] are both large-scale action recognition benchmarks, including 108k and 220k action clips, respectively. Both datasets share the same motion-centric action classes, *e.g.*, 'pushing something from left to right,' so thus capturing fine-grained motion is crucial to achieve the better performance.

**Diving-48** [30] is a fine-grained action benchmark that is heavily dependent on temporal modeling [3], containing 18k videos with 48 diving classes. Due to the incorrect label issue, we only compare our result with the results on the modified version of Diving-48.

**FineGym** [39] is a motion-centric benchmark that includes gymnastics action classes. We report results on two subsets of *Gym288* and *Gym99* that contain 288 and 99 action classes, respectively.

## 5.3 Comparison with other transform methods.

In this experiment, we evaluate the temporal modeling capability of different transform methods: spatio-temporal convolutions [48, 49, 50], self-attention [37] with its variants, and RSA. We replace a single $3 \times 3$ spatial convolution layer in TSN-ResNet50 [54] with a single spatio-temporal transform layer. We analyze the ability of modeling temporal dependency of each transform layer with an apple-to-apple comparison. We use 8 frames as the input, and the kernel size $M$ of all spatio-temporal transforms is set to 5×7×7.

Table 2 summarizes the results. 2D convolution baseline without modeling temporal dependency shows the lowest accuracy of 19.7%. While 3D [48] and (2+1)D [50] convolutions, which use static spatio-temporal transforms, show the top-1 accuracy of 43.3% and 44.1%, respectively, self-attention only achieves 41.6%. To find out the reasons behind the bad result, we ablate each component of self-attention, *e.g.*, content-to-content interaction ($\boldsymbol{x}_n^Q(\boldsymbol{X}_n^K)^\mathsf{T}$), content-to-position interaction ($\boldsymbol{x}_n^Q \boldsymbol{P}^\mathsf{T}$),

Table 3: **Performance comparison on SS-v1&v2, FineGym, and Diving-48**.

(a) **SS-V1&V2**. IN and IN21K and K400 denote ImageNet-1k, ImageNet-21K, and Kinetics-400 dataset, respectively. Our method achieves a new state-of-the-art accuracy on both datasets.

| model | pre-trained | #frame | FLOPs ×clips | SS-V1 top-1 | SS-V1 top-5 | SS-V2 top-1 | SS-V2 top-5 |
|---|---|---|---|---|---|---|---|
| I3D [5] from [56] | IN | 32 | 153 G×2 | 41.6 | 72.2 | - | - |
| TSM-R50 [31] | IN | 16 | 65 G×1 | 47.2 | 77.1 | 63.4 | 88.5 |
| ir-CSN-152 [49] | - | 32 | 97 G×10 | 49.3 | - | - | - |
| SlowFast8×8-R50 [11] | K400 | 32 | 67 G×3 | - | - | 61.7 | 86.9 |
| CT-Net-R50 [28] | IN | 16 | 75 G × 1 | 52.5 | 80.9 | 64.5 | 89.3 |
| STM-R50 [21] | IN | 16 | 67 G×30 | 50.7 | 80.4 | 64.2 | 89.8 |
| CorrNet-R101 [53] | - | 32 | 187 G×10 | 50.9 | - | - | - |
| TEA [29] | IN | 16 | 70 G ×3 | 52.3 | 81.9 | 65.1 | 89.9 |
| MSNet-TSM-R50 [23] | IN | 16 | 67 G×1 | 52.1 | 82.3 | 64.7 | 89.4 |
| NL-I3D [55] from [56] | IN | 32 | 168 G×2 | 44.4 | 76.0 | - | - |
| TimeSformer-HR [3] | IN | 16 | 1703 G×3 | - | - | 62.2 | - |
| TimeSformer-L [3] | IN | 96 | 2380 G×3 | - | - | 62.4 | - |
| ViViT-L [1] | IN21K & K400 | 32 | N/A×4 | - | - | 65.4 | 89.8 |
| RSANet-R50 (ours) | IN | 8 | 36 G×1 | 51.9 | 79.6 | 64.8 | 89.1 |
| RSANet-R50 (ours) | IN | 16 | 72 G×1 | 54.0 | 81.1 | 66.0 | 89.8 |
| RSANet-R50$_{EN}$ (ours) | IN | 8+16 | 108 G×1 | 55.5 | 82.6 | 67.3 | 90.8 |
| RSANet-R50$_{EN}$ (ours) | IN | 8+16 | 108 G×2 | **56.1** | **82.8** | **67.7** | **91.1** |

(b) **Diving-48**. Top-1 accuracy, FLOPs are shown. Results in the upper compartment are from [3].

| model | FLOPs ×clips | top-1 |
|---|---|---|
| SlowFast-R101 [11] | 213 G×3 | 77.6 |
| TimeSformer [3] | 196 G×3 | 75.0 |
| TimeSformer-HR [3] | 1703 G×3 | 78.0 |
| TimeSformer-L [3] | 2380 G×3 | 81.0 |
| RSANet-R50 | 72G×2 | **84.2** |

(c) **FineGym**. The averaged per-class accuracy (%) is reported. All results in the upper compartment are from [39].

| model | Gym288 | Gym99 |
|---|---|---|
| TRN [61] | 33.1 | 68.7 |
| I3D [5] | 27.9 | 63.2 |
| TSM [31] | 34.8 | 70.6 |
| TSM$_{Two\text{-}stream}$ [31] | 46.5 | 81.2 |
| RSANet-R50 | **50.9** | **86.4** |

and softmax ($\sigma$), one by one. We observe that the performance of self-attention without $x_n^Q P^\top$ significantly decreases, while that of self-attention without $x_n^Q (X_n^K)^\top$ is comparable to the original one. It indicates that the temporal modeling capability of the self-attention is actually dependent on the content-to-position interaction rather than the content-to-content interaction; the self-attention mechanism itself is permutation-invariant, so thus it is hard to learn position-specific features, *e.g.*, motion [32] without the positional embedding. After we remove the softmax non-linearity, where the kernel is equivalent to the basic kernel ($\kappa_n^V = x_n^Q P^\top$), outperforms both the self-attention and the standard static transforms. It demonstrates the softmax function restricts the expressive ability of the kernel [27]; the softmax forces the kernel weights to be positive, so that the kernel may not compute gradients across frames, which are effective in learning motion. Please refer to the Fig. 3 and Fig. 3 in our Supp. for the qualitative results.

While the composability of the basic kernel depends on the query content, our relational kernel depends on the local pattern of query-key correlation. As we add the relational kernel $\kappa_n^R$ to $\kappa_n^V$, we improve the top-1 accuracy by 1.4%p. The result demonstrates that leveraging local query-key interactions are effective in aggregating informative context for learning motion. At last, by adding relational context $X_n^R$ to basic context $X_n^V$, RSA achieves the best top-1 accuracy of 47.0%.

## 5.4 Comparison to the state-of-the-art methods

Table 3a compares our method to the state-of-the-art methods on SS-V1 and V2 datasets. The first compartment shows the results of the 3D CNNs. The second compartment contains the motion-modeling methods: both STM [21] and TEA [29] compute frame-wise differences, and both CorrNet [53] and MSNet [23] compute inter-frame pixel-wise correlations to learn motion features. The third compartment reports the results of global self-attention-based models. NL-I3D [55] inserts non-local blocks to the 3D CNN for capturing long-range spatio-temporal dependencies. TimeSformer-L, TimeSformer-HR [3] and ViViT [1] are transformer architectures that learn video representations via factorized spatio-temporal self-attention. Our method, RSANet-R50, achieves 51.9% and 64.8% at top-1 accuracy on SS-V1 and V2 datasets, respectively, which are already competitive to most of existing methods while using 8 frames only. When we use 16 frames, our method outperforms other existing models by achieving 54.0% and 66.0% at top-1 accuracy on SS-V1 and V2, respectively. Finally, our ensemble model with 2 clips achieves 56.1% and 67.7%, which set the new state-of-the-art on SS-V1 and V2 with reasonable computational cost.

Table 3c and Table 3b present the results on Diving-48 [30] and FineGym [39]. For Diving-48, our model achieves 84.2%, substantially outperforming the state-of-the-art 3D CNN and transformer

Table 4: **Ablation studies on SS-v1 dataset**. All models use TSN-ResNet50 [54] as the backbone. Top-1, top-5 accuracy (%), FLOPs (G) and paramaters (M) are shown.

(a) **Combinations of different kernels and context**. A single RSA layer is inserted into stage4.

| kernel | context | FLOPs | params. | top-1 | top-5 |
|---|---|---|---|---|---|
| $\kappa_n^{\mathrm{V}}$ | $X_n^{\mathrm{V}}$ | 32.3 G | 23.4 M | 44.8 | 73.8 |
| $\kappa_n^{\mathrm{R}}$ | $X_n^{\mathrm{V}}$ | 32.7 G | 23.6 M | 45.4 | 73.9 |
| $\kappa_n^{\mathrm{V}} + \kappa_n^{\mathrm{R}}$ | $X_n^{\mathrm{V}}$ | 32.7 G | 23.6 M | 45.7 | 74.8 |
| $\kappa_n^{\mathrm{V}}$ | $X_n^{\mathrm{R}}$ | 32.7 G | 23.4 M | 46.2 | 75.4 |
| $\kappa_n^{\mathrm{R}}$ | $X_n^{\mathrm{R}}$ | 33.2 G | 23.6 M | 46.5 | 75.6 |
| $\kappa_n^{\mathrm{V}} + \kappa_n^{\mathrm{R}}$ | $X_n^{\mathrm{R}}$ | 33.2 G | 23.6 M | 46.7 | 75.6 |
| $\kappa_n^{\mathrm{V}}$ | $X_n^{\mathrm{V}} + X_n^{\mathrm{R}}$ | 32.7 G | 23.4 M | 46.5 | 75.6 |
| $\kappa_n^{\mathrm{R}}$ | $X_n^{\mathrm{V}} + X_n^{\mathrm{R}}$ | 33.2 G | 23.6 M | 46.8 | 75.6 |
| $\kappa_n^{\mathrm{V}} + \kappa_n^{\mathrm{R}}$ | $X_n^{\mathrm{V}} + X_n^{\mathrm{R}}$ | 33.2 G | 23.6 M | **47.0** | **75.7** |

(b) **Latent dimension** $D$. Decomposing $H$ significantly reduces the computation cost. OOM is an abbreviation of out-of-memory. 8 video clips per a single GPU machine are used.

| $D$ | FLOPs | params. | memory | top-1 | top-5 |
|---|---|---|---|---|---|
| - | 62.9 G | 32.0 M | OOM | OOM | OOM |
| 8 | 32.2 G | 20.5 M | 8.8 GB | 50.1 | 78.8 |
| 16 | 34.7 G | 20.9 M | 9.2 GB | 51.3 | 78.8 |
| 32 | 39.6 G | 21.7 M | 10.2 GB | 50.9 | 79.0 |
| $C^{\mathrm{Q}}/2$ | 32.9 G | 21.1 M | 8.8 GB | 51.1 | 79.1 |
| $C^{\mathrm{Q}}$ | 35.9 G | 22.0 M | 9.6 GB | **51.5** | **79.2** |

(c) **Kernel size** $M$. In most cases, larger kernel results in the higher accuracy.

| kernel size $M$ | FLOPs | params. | top-1 | top-5 |
|---|---|---|---|---|
| $3 \times 3 \times 3$ | 28.5 G | 20.3 M | 49.4 | 77.6 |
| $3 \times 5 \times 5$ | 30.2 G | 20.7 M | 50.5 | 78.7 |
| $3 \times 7 \times 7$ | 32.6 G | 21.2 M | 50.7 | 78.9 |
| $3 \times 9 \times 9$ | 35.8 G | 22.0 M | 51.1 | 79.1 |
| $5 \times 7 \times 7$ | 35.9 G | 22.0 M | **51.5** | **79.2** |
| $5 \times 9 \times 9$ | 41.3 G | 23.3 M | 51.2 | 78.9 |

(d) **Group** $G$. Hadamard product ($G = C^{\mathrm{Q}}$) performs the highest accuracy. Note that FLOPs are consistent with varying $G$ due to the switched computation order.

| # Groups $G$ | FLOPs | params. | top-1 | top-5 |
|---|---|---|---|---|
| 1 | 35.9 G | 20.2 M | 50.4 | 78.9 |
| 2 | 35.9 G | 20.2 M | 50.9 | 78.9 |
| 4 | 35.9 G | 20.3 M | 51.2 | 78.9 |
| 8 | 35.9 G | 20.5 M | 51.2 | 79.0 |
| $C^{\mathrm{Q}}$ | 35.9 G | 22.0 M | **51.5** | **79.2** |

architectures. For FineGym, our model outperforms other methods in the averaged per-class accuracy of 50.9% and 86.4% given 288 and 99 classes, respectively.

## 5.5 Ablation studies

We conduct ablation experiments to validate the effectiveness of RSA. We use 8 frames for all experiments. Other training and testing details are in Sec. 5.1.

**Combinations of different kernels and contexts.** In Table 4a, we compare the performance of a single RSA layer with different combinations of dynamic kernels and contexts. We first vary different types of the kernels from $\kappa_n^{\mathrm{V}}$, $\kappa_n^{\mathrm{R}}$ to $\kappa_n^{\mathrm{V}} + \kappa_n^{\mathrm{R}}$, while the context is fixed as $X_n^{\mathrm{V}}$. Compared to $\kappa_n^{\mathrm{V}} X_n^{\mathrm{V}}$, $\kappa_n^{\mathrm{R}} X_n^{\mathrm{V}}$ improves the accuracy by 0.6%. It indicates that predicting kernel from the composition of local query-key correlation is effective in modeling temporal dependencies. As we use both of the basic and relational transforms, $(\kappa_n^{\mathrm{V}} + \kappa_n^{\mathrm{R}}) X_n^{\mathrm{V}}$, we obtain additional improvements by 0.3%p and 0.9%p at top-1 and top-5 accuracy, respectively. We also vary different types of the context such as $X_n^{\mathrm{V}}$, $X_n^{\mathrm{R}}$, and $X_n^{\mathrm{V}} + X_n^{\mathrm{R}}$, while the kernel is fixed. We observe the consistent improvements across different types of contexts, which means that aggregating the relational context is beneficial for learning relational information. Finally, as we combine all dynamic transforms, $(\kappa_n^{\mathrm{V}} + \kappa_n^{\mathrm{R}})(X_n^{\mathrm{V}} + X_n^{\mathrm{R}})$, we achieve the highest accuracy of 47.0%.

**Latent dimension** $D$**.** In Table 4b, we validate the effectiveness of decomposing $H$ with latent channel dimension $D$. Without decomposing $H$, the TSN-ResNet50 with 7 RSA layers requires 62.9 GFLOPs and 32.0 M parameters, resulting in out-of-memory error. After we decompose $H$ with a small $D$, we significantly reduce FLOPs, the number of parameters, and the memory footprint. We set $D = C^{\mathrm{Q}}$ as the default that performs the highest accuracy.

**Kernel sizes.** In Table 4c, we compare the effect of the kernel size $M$. In most cases, larger spatio-temporal kernel improves the accuracy, except the case of $M = 5 \times 9 \times 9$. Considering the static convolution counterparts, the RSA effectively enlarges the spatio-temporal kernel size, requiring smaller FLOPs and parameters. We choose $M = 5 \times 7 \times 7$ as the default kernel size that shows the best computation-accuracy trade-off.

**Correlation computation.** In Table 4d, we validate the effect of the group-wise correlation [15, 58], which splits the query and the key embeddings into $G$ groups and computes a dot product between each group. While the dot product correlation contracts all channel dimensions into a scalar correlation

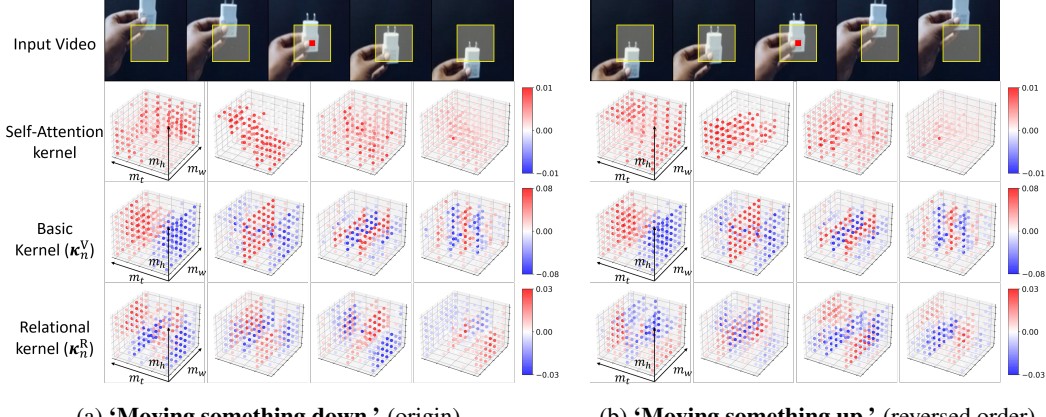

Figure 3: **Kernel visualization results on SS-V1**. From the top to the bottom in each subfigure, we visualize the input RGB frames, the self-attention kernels, the basic kernels, and the relational kernels. The query position and the context are marked as red and yellow in RGB frames, respectively. The size of spatio-temporal kernel $M$ is set as $m_t \times m_h \times m_w = 5 \times 7 \times 7$ and 4 kernels out of $L$ kernels ($L = 8$) are shown for each transform.

score as the dot-product self-attention [52], the group-wise correlation outputs a $G$-dimensional correlation vector, providing richer semantic information. As a result, the Hadamard product, *i.e.*, $G = C^Q$, that computes full element-wise correlations achieves the best performance. We thus set the Hadamard product as the default correlation function. Note that the computational complexity for generating the relational kernel remains the same while the size of $G$ varies since we switch the orders of computation as in Eq. 10.

### 5.6 Kernel Visualization

In Figure 3, we visualize the dynamic kernels of self-attention and RSA from the learned models in $4^{\text{th}}$ and $7^{\text{th}}$ rows in Table 3, respectively. In visualization, we observe that both of the basic kernels and relational kernels resemble edge detectors, *e.g.*, Sobel filters or Laplacian filters [46], along a temporal axis that compute a discrete approximation of spatio-temporal gradients across frames. When we reverse the temporal order of an input video clip, the relational kernel dynamically varies according to whether the object moves up or down but the basic kernel remains the same. Considering that motion information is related to the relative changes along the temporal axis, the results indicate that the RSA kernels effectively capture motion patterns within the context, whereas the self-attention kernels are limited to aggregating the local context based on the query-key similarities.

## 6  Conclusion

We have presented the RSA feature transform, which captures rich relational patterns for video understanding, and validated that it outperforms other dynamic feature transforms in learning motion dynamics in videos. The proposed RSANet outperforms the state-of-the art methods on standard motion-centric action benchmarks. While we have focused on video representation learning in this work, we believe the RSA will also benefit image understanding and natural language processing. We leave this for future work.

## Acknowledgments and Disclosure of Funding

This work was supported by the NRF grants (NRF-2017R1E1A1A01077999, NRF-2021R1A2C3012728), and the IITP grant (No.2019-0-01906, AI Graduate School Program - POSTECH) funded by Ministry of Science and ICT, Korea. This work was done while Manjin was working as an intern at Microsoft Research Asia.

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
