# Relational Self-Attention: What's Missing in Attention for Video Understanding
## *Supplementary Material*

**Manjin Kim**[1*]   **Heeseung Kwon**[1*]   **Chunyu Wang**[2]   **Suha Kwak**[1]   **Minsu Cho**[1]

[1]POSTECH        [2]Microsoft Research Asia

http://cvlab.postech.ac.kr/research/RSA/

## 1   Implementation details

**Architecture details.** We use TSN-ResNet [11] as our backbone (see Table 1) and initialize it with ImageNet-pretrained weights [4]. We replace its 7 spatial convolutional layers with the RSA layers; for every two ResNet blocks from the third block in $res_2$ to the second block in $res_5$, each spatial convolutional layer is replaced with the RSA layer.

Table 1: **TSN-ResNet-50 backbone**.

| Layers | TSN-ResNet-50 | Output size |
|---|---|---|
| $conv_1$ | $1{\times}7{\times}7$, 64, stride 1,2,2 | $T{\times}112{\times}112$ |
| $pool_1$ | $1{\times}3{\times}3$ max pool, stride 1,2,2 | $T{\times}56{\times}56$ |
| $res_2$ | $\begin{bmatrix} 1{\times}1{\times}1,\ 256 \\ 1{\times}3{\times}3,\ 64 \\ 1{\times}1{\times}1,\ 256 \end{bmatrix} \times 3$ | $T{\times}56{\times}56$ |
| $res_3$ | $\begin{bmatrix} 1{\times}1{\times}1,\ 512 \\ 1{\times}3{\times}3,\ 128 \\ 1{\times}1{\times}1,\ 512 \end{bmatrix} \times 4$ | $T{\times}28{\times}28$ |
| $res_4$ | $\begin{bmatrix} 1{\times}1{\times}1,\ 1024 \\ 1{\times}3{\times}3,\ 256 \\ 1{\times}1{\times}1,\ 1024 \end{bmatrix} \times 6$ | $T{\times}14{\times}14$ |
| $res_5$ | $\begin{bmatrix} 1{\times}1{\times}1,\ 2048 \\ 1{\times}3{\times}3,\ 512 \\ 1{\times}1{\times}1,\ 2048 \end{bmatrix} \times 3$ | $T{\times}7{\times}7$ |
| global average pool, FC | | # of classes |

**Training.** For the bottlenecks including RSA layers, we randomly initialize weights using MSRA initialization [3] and set the gamma parameter of the last batch normalization layer to zero. We sample 8 or 16 frames from each video by using a segment-based random sampling strategy [11]. We resize the resolution of each frame to $240 \times 320$, and apply random cropping as $224 \times 224$, scale jittering, and random horizontal flipping for data augmentation. Note that we do not flip videos of which action labels include 'left' or 'right' words, *e.g.*, 'pulling something from left to right'. We set the initial learning rate to 0.02 and decay it with a cosine learning rate [7] after the initial 5 warm-up epochs. We use SGD with the momentum of 0.9 and set the batch size as 64 across 8 V100 GPU machines. For training SS-V1&V2 [2], we train models for 50 epochs in total except that the models in Table 2a in our main paper are trained for 80 epochs. We use dropout of 0.3 before the final classifier, label smoothing [9] of 0.1 and use stochastic depth [5] with rate 0.2 for regularization. For Diving-48 [6], we sample 16 frames from each video and train for total 30 epochs. We use dropout of 0.5 before the final classifier for regularization. For FineGym [8], we sample 8 frames from each video and train for total 50 epochs. We use dropout of 0.5 before the final classifier.

---

[*]Equal contribution.

35th Conference on Neural Information Processing Systems (NeurIPS 2021).

```
# B: batch size, N: input size, M: kernel size, C: channel dim, D: latent dim
def compute_relational_kernel(query, key, H1, H2, impl='Eq_10'):
# query shape: [B,N,C], key shape: [B,N,C], H1 shape: [MC,D], H2 shape: [M,D]
    if impl == 'Eq_9':
        key = unfold(key, unfold_size=M) # key shape: [B,N,M,C]
        qk_Hadamard = einsum(query, key, 'BNC,BNMC->BNMC') # shape: [B,N,M,C]
        qk_H1 = einsum(qk_Hadamard, H1, 'BNMC,MCD->BND') # shape: [B,N,D]
    elif impl == 'Eq_10':
        H1 = reshape(H1, [M,CD]) # shape: [M,CD]
        key_H1 = conv1d(key, H1, group=C, kernel=M, bias=False) # shape: [B,N,CD]
        key_H1 = reshape(key_H1, [B,N,C,D]) # shape: [B,N,C,D]
        qk_H1 = einsum(query, key_H1, 'BNC,BNCD->BND') # shape: [B,N,D]
    relational_kernel = einsum(qk_H1,H2, 'BND,MD->BNM') # shape: [B,N,M]
    return relational_kernel
```

Figure 1: **Pseudo-code for the relational kernel.** By switching the computation orders, we can significantly reduce the memory footprint of the relational kernel.

**Testing.** For all benchmarks, we sample one or two clips, resize them to $240 \times 320$, and center-crop by $224 \times 224$. For Diving-48 [6], we sample two clips consist of 16 frames, and average softmax scores for inference. For FineGym [8], we sample a single clip consists of 8 frames for inference.

**License.** We implement our model based on TSN in Pytorch[2] under BSD 2-Clause license. All the benchmarks that we used are commonly used datasets for the academic purpose. According to https://paperswithcode.com/datasets, the license of SS-V1&V2 and FineGym is the custom[3] and the CC BY-NC 4.0, respectively, and the license of Diving-48 is unknown.

**Implementation of the relational kernel.** Generating relational kernels spends a large amount of memory, and it thus requires CUDA implementations to be optimized. As described in Sec.4.2, we switch the computation orders, and then dramatically reduce the memory footprint. In Figure 1, we present pseudo-codes of Eq.9 and 10 in our main paper for generating the relational kernel, and notate the size of the intermediate tensors. As illustrated in the pseudo-code, the memory footprint is reduced from $\mathcal{O}(BNMC)$ to $\mathcal{O}(BNC^2)$, where $D = C$ and $C$ is much smaller than $M$ in our experiments. For ease description, the notation of multi-query $L$ is omitted.

**Implementation of RSA.** In Figure 2, We provide pseudo-codes of Eq.11 and 12 in Sec.4.2 in our main paper for computing RSA and notate the sizes of intermediate tensors. We can substantially reduce the memory by decomposing $\boldsymbol{P}$ and our RSA can be easily implemented with convolutions and simple einsum operations. As illustrated in Figure 2, the memory footprint is reduced from $\mathcal{O}(BNMC + MC^2)$ to $\mathcal{O}(BNC^2 + MC^2)$, where $D = C$ and $C$ is much smaller than $M$ in our experiments. For ease description, the notation of multi-query $L$ is omitted.

## 2 Additional ablation studies on RSA.

In this section, we provide additional ablation experiments on SS-V1 to validate the effect of other design components in the RSA. While specified otherwise, the training and testing details are the same as those in Sec.5.1.

**Multi-query RSA.** In Table 2a, we investigate the effect of multi-query RSA by varying the number of queries $L$. The table shows that using more queries substantially reduces the computational cost of the RSA as shown in Table 1 in our main paper. Despite the channels of $\boldsymbol{x}_n^{\mathrm{Q}}, \boldsymbol{X}_n^{\mathrm{K}}, \boldsymbol{X}_n^{\mathrm{V}}$ are reduced to $C/L$ as $L$ increased, the top-1 accuracy is slightly improved until $L = 8$. Since each RSA kernel generated by each query captures a distinct motion pattern, the model can learn diverse motion features (see Figure 3). In this experiment, we choose $L = 8$ as the default.

**Number and locations of RSA layers.** In Table 2b, we gradually replace the spatial convolutional layers with RSA layers. Adding a single RSA layer significantly improves the top-1 accuracy by 27.3%p. As we increase the number of RSA layers, we obtain gradual improvements of accuracy, while reducing parameters only. Note that RSA layer with the smaller kernel size $M$, *e.g.*, $3 \times 5 \times 5$,

[2]URL: https://github.com/yjxiong/tsn-pytorch
[3]URL: https://20bn.com/licensing/datasets/academic

```
# B: batch size, N: input size, M: context size, C: channel dim, D: latent dim
def RSA(query, key, value, P, H1, H2, G, impl='Eq_12'):
# query shape: [B,N,C], key shape: [B,N,C], value: [B,N,C],
# P1 shape: [M,D], H1 shape: [MC,D], H2 shape: [M,D], G shape: [M,C]
    if impl == 'Eq_11':
        # kernels
        basic_kernel = einsum(query, P, 'BNC,CM->BNM') # shape: [B,N,M]
        H1 = reshape(H1, [M,CD]) # shape: [M,CD]
        key_H1 = conv1d(key, H1, group=C, kernel=M, bias=False) # shape: [B,N,CD]
        key_H1 = reshape(key_H1, [B,N,C,D]) # shape: [B,N,C,D]
        qk_H1 = einsum(query, key_H1, 'BNC,BNCD->BND') # shape: [B,N,D]
        relational_kernel = einsum(qk_H1, H2, 'BND,MD->BNM') # shape: [B,N,M]
        kernel = basic_kernel + relational_kernel # shape: [B,N,M]
        # contexts
        value = unfold(value, unfold_size=M) # value shape: [B,N,M,C]
        v_G = einsum(value, G, 'BNMC,MC->BNCC') # shape: [B,N,C,C]
        relational_context = einsum(value, v_G, 'BNMC,BNCC->BNMC') # shape: [B,N,M,C]
        context = value + relational_context # shape: [B,N,M,C]
        RSA = einsum(kernel, context, 'BNM,BNMC->BNC') # shape: [B,N,C]
    elif impl == 'Eq_12':
        P1_k_H1 = conv1d(key,H1, group=C, kernel=M, bias=P1) # shape: [B,N,CD]
        P1_k_H1 = reshape(P1_k_H1, [B,N,C,D]) # shape: [B,N,C,D]
        q_P1_k_H1 = einsum(query, P1_k_H1, 'BNC,BNCD->BND') # shape: [B,N,D]
        value = reshape(value, [B,NC,1]) # shape: [B,NC,1]
        H2_v = conv1d(value, H2, kernel=M, bias=False) # shape: [B,NC,D]
        H2_v = reshape(H2_v, [B,N,C,D]) # shape: [B,N,C,D]
        I_v_G = conv1d(value, G, kernel=M, bias=I) # shape: [B,NC,C]
        I_v_G = reshape(I_v_G, [B,N,C,C]) # shape: [B,N,C,C]
        H2_v_I_v_G = einsum(H2_v, I_v_G, 'BNCD,BNCC->BNCD') # shape: [B,N,C,D]
        RSA = einsum(q_P1_k_H1, H2_v_I_v_G, 'BND,BNCD->BNC') # shape: [B,N,C]
    return RSA
```

Figure 2: **Pseudo-code for the RSA.** By switching the Eq.11 to the Eq.12, we can significantly reduce the memory footprint of RSA.

requires smaller amount of both FLOPs and parameters, while covering larger receptive fields than the convolution as shown in the second row of Table 4c in our main paper.

**Normalization.** In Table 2c, we investigate different normalization methods to $x_n^{\mathrm{Q}}$, $X_n^{\mathrm{K}}$, and $X_n^{\mathrm{V}}$. In our setting, applying L2 normalization performs the best. Notably, applying no normalization perform better than applying batch normalization or normalization scheme from [1] due to the ImageNet-pre-trained weights.

# 3 Kernel visualization

We present additional visualization results of RSA kernels. In Figure 3, we visualize the dynamic kernels of self-attention and RSA from the learned models in 4th and 7th rows in Table 2 in the main paper, respectively. In visualization, we observe that each RSA kernel from each query captures a distinct motion pattern, and its values are dynamically modulated by the input query and the context. For example, three RSA kernels at the bottom row of each subfigure are noticeably modulated by the spatio-temporal correlations between the query and the context. It demonstrates that the RSA kernel determines its composability based on the input instance, leading to flexible video representation learning.

# 4 Potential societal impacts

Video action recognition is a fundamental task in the computer vision, and our work thus has a potential to assist numerous application scenarios, such as video retrieval, video surveillance, autonomous driving, and sports analytics, etc. However, the research on the video domain can be used for nefarious purposes, especially in the area of unauthorized surveillance. We believe the positive

Table 2: **Additional ablation studies on SS-v1 dataset**. All models use TSN-ResNet50 [11] as the backbone. Top-1, top-5 accuracy (%), FLOPs (G) and paramaters (M) are shown.

(a) **Query** $L$. Using multiple queries reduces the computational cost. Latent dimension $D$ is set to 16. OOM is an abbrebiation of out-of-memory.

| # query $L$ | FLOPs | params. | top-1 | top-5 |
|---|---|---|---|---|
| 1 | 252.5 G | 28.1 M | OOM | OOM |
| 2 | 92.8 G | 23.9 M | 50.6 | 78.9 |
| 4 | 48.2 G | 21.9 M | 50.9 | **79.0** |
| 8 | 34.7 G | 20.9 M | **51.3** | 78.8 |
| 16 | 30.2 G | 20.4 M | 50.0 | 78.2 |

(b) **Number and locations of RSA layers**. More RSA layers lead the better performance.

| number | positions | FLOPs | params. | top-1 | top-5 |
|---|---|---|---|---|---|
| 0 | - | 32.5 G | 24.3 M | 19.7 | 46.6 |
| 1 | stage4 | 33.2 G | 23.6 M | 47.0 | 75.7 |
| 2 | stage4-5 | 33.7 G | 22.6 M | 47.2 | 76.1 |
| 4 | stage4-5 | 34.6 G | 22.2 M | 50.4 | 78.7 |
| 7 | stage2-5 | 35.9 G | 22.0 M | **51.5** | **79.2** |

(c) **Normalization**. L2-normalization performs the best accuracy in our setting.

| normalization | FLOPs | params. | top-1 | top-5 |
|---|---|---|---|---|
| no norm. | 35.9 G | 22.0 M | 50.3 | 78.6 |
| batch norm. | 35.9 G | 22.0 M | 49.4 | 78.0 |
| norm. from [1] | 35.8 G | 22.0 M | 49.6 | 78.2 |
| L2-norm. | 35.9 G | 22.0 M | **51.5** | **79.2** |

impacts of our work significantly outweigh the harmful impacts, but we clarify that it is dependent to the purpose of the users.

## 5 Limitations

The effectiveness of the RSA transform has been demonstrated through our work, but it still leaves much room for improvement.

- First, the computational efficiency of the RSA could be further improved. For example, decomposing a spatio-temporal 3D RSA kernel into a spatial 2D and a temporal 1D kernel [10] could reduce the computational complexity.

- Second, RSA captures both visual appearance and relational features with an additive manner, but we anticipate that there will be a more generalized dynamic transform that captures both features in natural. We hope that RSA could be utilized as a guideline for designing the generalized dynamic transform.

- Third, we believe that leveraging relational information can also benefit task of different domains such as image understanding or natural language processing. We leave this for future work.

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

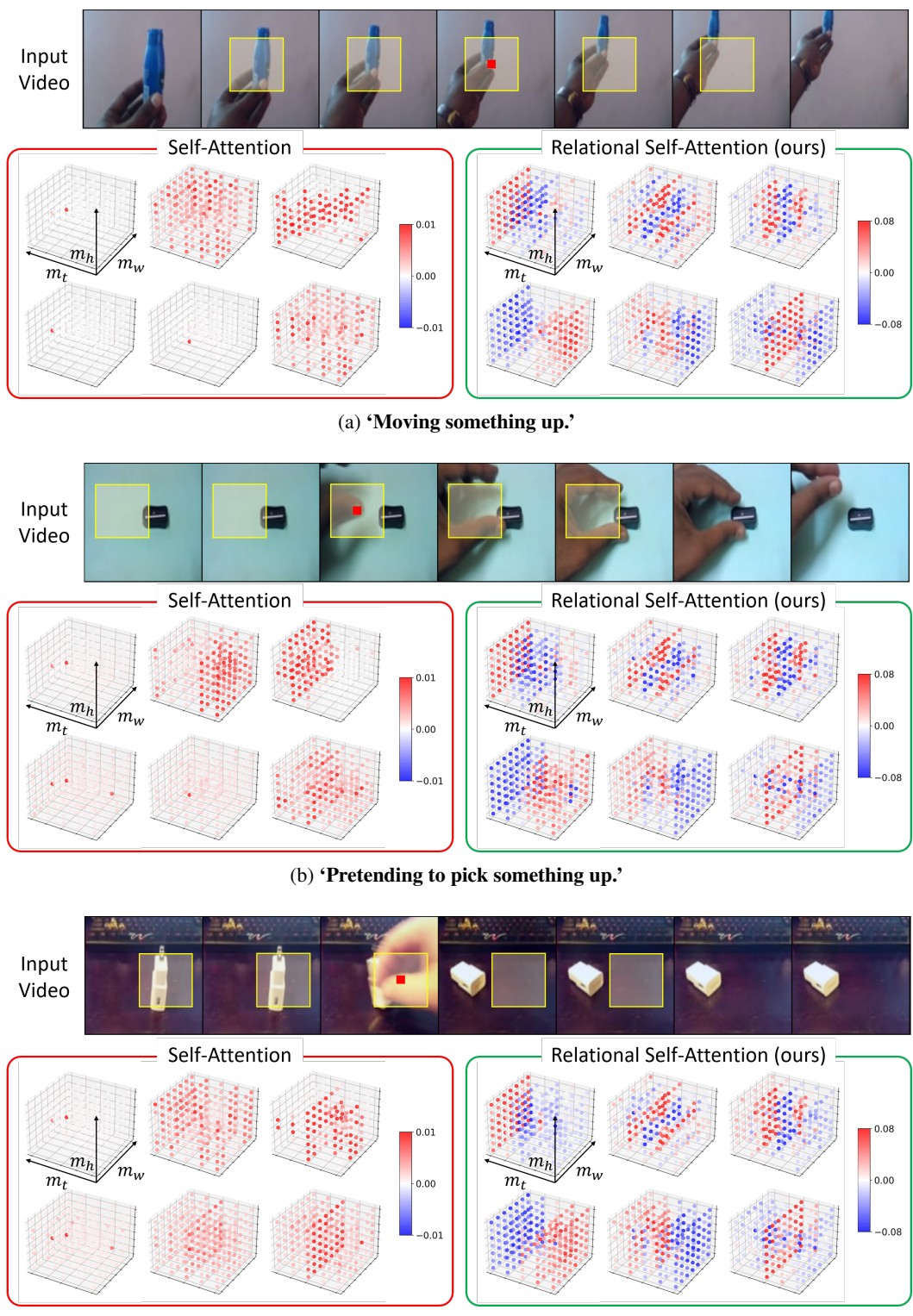

(a) 'Moving something up.'

(b) 'Pretending to pick something up.'

(c) 'Throwing something in the air and catching it.'

Figure 3: **Kernel visualization results on SS-V1**. In each subfigure, we visualize the input RGB frames (top), the self-attention kernels (left), and the RSA kernels (right). The query position and the context are marked as red and yellow in RGB frames, respectively. The size of spatio-temporal kernel $M$ is set as $m_t \times m_h \times m_w = 5 \times 7 \times 7$ and 6 kernels out of $L$ kernels ($L = 8$) are shown for each transform.