# OpenReview forum: "Relational Self-Attention: What's Missing in Attention for Video Understanding"
_NeurIPS.cc/2021/Conference — NeurIPS 2021 Poster_

### Official Review · Reviewer_iQAh · 2021-07-16

**Rating:** 6
**Confidence:** 4

**Summary:**

This paper proposes a new transform block called Intervolution for video understanding. Specifically, the Intervolution block is based on the recent dynamic transform blocks and further models the content-to-content relations in video data. The new designs of relational kernel and relational context improve the ability of dynamic transform to capture spatio-temporal relations. The experiments are conducted on three public datasets, i.e., SS-V1&V2, FineGym, and Diving-48. The experimental results show that the networks with Intervolution achieve higher accuracy on action recognition than other approaches.

**Limitations And Societal Impact:**

The authors have discussed the limitations of this work. The potential negative societal impact is not discussed.

**Main Review:**

Strengths:
+ The paper is mostly clear and easy to follow.
+ The utilization of dynamic transform on video understanding is interesting.
+ The performances are high on public datasets.


Weaknesses:
- Why Intervolution for video data. The design of Intervolution is based the dynamic transforms for image understanding. And the architecture of Intervolution is not specifically designed for video data. I am not sure why this paper only validate Intervolution on video recognition task. Although the paper claims that the relation modelling in Intervolution could help capture motion information in video, I think relation is also important for image understanding. It is not a convincing reason why the paper only experimented on video data. Meanwhile, without the results on image data, it is hard to directly compare Intervolution with other dynamic transforms.
- Comparison with Involution and Lambda convolution. To me, the architecture of Intervolution combines the several designs in self-attention, Involution and Lambda convolution. However, in Table 2, only self-attention (LSA) is compared. It is not surprising that Intervolution is higher than LSA since Involution and Lambda convolution have already been proven to be better than LSA.  I understand that some settings in the ablation study are similar as Involution or Lambda convolution. The more direct comparisons with these two blocks should be given.
- Dataset. The experiments are mainly conducted on the motion-centric datasets. It is acceptable since these datasets could better validate the ability of Intervolution to capture motion information. However, I still wonder how Intervolution performs on the datasets focusing more on object/scene (e.g., Kinetics dataset). The authors could give some results on such datasets or discuss these datasets to further provide some valuable insights on dataset bias.
- Backbone. The experiments in this paper mainly utilize ResNet50 as backbone. The results on more backbones could make the conclusions more convincing.


**Time Spent Reviewing:**

10

---

> ### Author Response · Authors · 2021-08-10
> **Response to Reviewer iQAh**
>
> Thanks for your constructive comments.
>
> ***
>
> > [R iQAh] Why Intervolution for video data. The design of Intervolution is based on dynamic transforms for image understanding. And the architecture of Intervolution is not specifically designed for video data. I am not sure why this paper only validates Intervolution on video recognition tasks. Although the paper claims that the relation modelling in Intervolution could help capture motion information in video, I think relation is also important for image understanding. It is not a convincing reason why the paper only experimented on video data. Meanwhile, without the results on image data, it is hard to directly compare Intervolution with other dynamic transforms.
>
> Thanks for the constructive comment. We have chosen to focus on the video domain because the motivation of this work comes from the fact that recent dynamic feature transforms have failed to improve [A1,A2] or have not been evaluated [A3,A4] on the video domain (L27-34). However, we fully agree with the reviewer’s point that Intervolution is a generic feature transform method so that it can improve on other tasks, in particular, image classification. Following your suggestion, we will seriously consider including the results on image classification and revising the manuscript accordingly. This may also help interest more readers.
>
> As the reviewer requested, we conducted additional experiments on image classification. Here, due to the lack of time and GPU resources, we experimented on CIFAR-10 & -100 only. Table A below summarizes the results. For a fair comparison, we use the same hyper-parameters of dynamic transforms such as the number of queries or channels, and control the FLOPs of Intervolution to match those of the convolution. The results show that Intervolution outperforms the convolution on both CIFAR-10 & -100 with fewer parameters, while existing dynamic transforms perform similar to convolution on CIFAR-10 & 100. It validates that leveraging rich relational information is also helpful in image recognition. We will conduct the same experiments on large-scale benchmarks, *e.g.*, ImageNet, and consider including them in the final version.
>
>
> |model (ResNet-20)		|FLOPs (M)	|Params (K)	| CIFAR-10 Top-1 err (%)	| CIFAR-100 Top-1 err (%)|
> |--|--|--|--|--|
> |w/ Convolution		|65.2		|458.5		|8.1		|29.5|
> |w/ Local Self-Attention	|57.1		|349.3		|8.1		|30.8|
> |w/ Lambda convolution	|55.9		|341.2		|7.8		|29.5
> |w/ Intervolution		|65.4		|362.4		|**7.3**		|**28.5**|
>
> Table A: Performance comparison with other spatial feature transform methods on CIFAR-10 and CIFAR-100. FLOPs and # Params are measured when CIFAR-100 is used.
>
> In the following, we provide implementation details for our experiments. We adopt ResNet-20, which consists of 6 bottlenecks, as the backbone and replace three $3\times3$ conv layers by three feature transform layers whose size of spatial context is set to $7\times7$. Note that We set the latent dimension D of Intervolution to 4. We follow training & testing protocols in CutMix paper [A5]. We train the models from scratch and set the initial learning rate to 0.1, and then decay it by 1/10 after 100th and 150th epochs. We set total epochs to 200.
>
> [A1] A. Arnab *et al*., “ViViT: A Video Vision Transformer,” arXiv, 2021.\
> [A2] G. Bertasius *et al*., “Is Space-Time Attention All You Need for Video Understanding?” arXiv, 2021.\
> [A3] D. Li *et al*., “Involution: Inverting the Inherence of Convolution for Visual Recognition,” CVPR, 2021.\
> [A4] I. Bello, “LambdaNetworks: Modeling Long-Range Interactions Without Attention,” ICLR, 2021.\
> [A5] S. Yun *et al*., “CutMix: Regularization Strategy to Train Strong Classifiers with Localizable Features,” ICCV, 2019.
>
> ***
>
> > [R iQAh] Comparison with Involution and Lambda convolution. To me, the architecture of Intervolution combines the several designs in self-attention, Involution and Lambda convolution. However, in Table 2, only self-attention (LSA) is compared. It is not surprising that Intervolution is higher than LSA since Involution and Lambda convolution have already been proven to be better than LSA. I understand that some settings in the ablation study are similar as Involution or Lambda convolution. The more direct comparisons with these two blocks should be given.
>
> The direct comparison with Lambda convolution is already presented in Table 2 of the manuscript; the LSA variant in 6th row of Table 2 is equivalent to Eq.4 of spatio-temporal Lambda convolution. Since the only difference between Involution (Eq.3) and Lambda convolution (Eq.4) is the existence of the two learnable embeddings, $\boldsymbol{E}^\mathrm{Q}$ and $\boldsymbol{E}^\mathrm{V}$, we conducted an additional experiment to compare with Involution by removing learnable embedding layers from the Lambda convolution. The result is shown in Table B below. Involution performs worse than Lambda convolution by 0.9%p. Compared to Involution and Lambda convolution, Intervolution with relational kernels outperforms them by ~1%p at top-1 accuracy, showing the effectiveness of the relational kernels.
>
> |model (TSN-R50)|kernel	|context|FLOPs (G)|Params (M)|top-1 (%)	|top-5 (%)|
> |--|--|--|--|--|--|--|
> |w/ LSA			| $\sigma(\boldsymbol{x}_n^{\mathrm{Q}}(\boldsymbol{X}_n^{\mathrm{K}})^\top + \boldsymbol{x}_n^{\mathrm{Q}} \boldsymbol{P}^\top)$	|$\boldsymbol{X}_n^{\mathrm{V}}$	|32.2		|24.3		|41.6	|70.9|
> |w/ Lambda convolution |$\boldsymbol{x}_n^{\mathrm{Q}}(\boldsymbol{X}_n^{\mathrm{K}})^\top$		|$\boldsymbol{X}_n^{\mathrm{V}}$	|32.2		|23.4		|44.3	|73.7|
> |w/ Involution		|$\boldsymbol{x}_n(\boldsymbol{X}_n^{\mathrm{K}})^\top$		|$\boldsymbol{X}_n$	|32.0		|23.3		|43.2	|72.2|
> |w/ intervolution	|$\boldsymbol{\kappa}_n^{\mathrm{R}}$		|$\boldsymbol{X}_n^{\mathrm{V}}$	|32.6		|23.4		|45.4	|73.9|
> |w/ intervolution	|$\boldsymbol{\kappa}_n^{\mathrm{V}} + \boldsymbol{\kappa}_n^{\mathrm{R}}$	|$\boldsymbol{X}_n^{\mathrm{V}}$	|32.6		|23.4		|45.7	|74.8|
> |w/ intervolution	|$\boldsymbol{\kappa}_n^{\mathrm{V}} + \boldsymbol{\kappa}_n^{\mathrm{R}}$	|$\boldsymbol{X}_n^{\mathrm{V}} + \boldsymbol{X}_n^{\mathrm{R}}$	|33.2		|23.6		|**47.0**|**75.7**|
>
> Table B: Performance comparison with other spatio-temporal feature transform methods on SS-v1.
>
> ***
>
>
> > [R iQAh] Dataset. The experiments are mainly conducted on the motion-centric datasets. It is acceptable since these datasets could better validate the ability of Intervolution to capture motion information. However, I still wonder how Intervolution performs on the datasets focusing more on object/scene (e.g., Kinetics dataset). The authors could give some results on such datasets or discuss these datasets to further provide some valuable insights on dataset bias.
>
> Although we evaluate Intervolution on motion-centric datasets, Intervolution is a generic dynamic feature transform that could be readily applied to any existing video datasets. To validate the versatility of Intervolution, we conducted an additional experiment on Kinetics-400 dataset, which is one of the appearance-centric action benchmarks. Table C below summarizes the results, which show that Intervolution outperforms other feature transform methods when the same size of the local context is used. We validate that the relational kernel and the relational context still show a clear gain on the appearance-centric benchmark.
>
> |model (TSN-R50)	|FLOPs (G)	|Params (M)	|top-1 (%)	|top-5 (%)|
> |--|--|--|--|--|
> |w/ R(2+1)D		|37.3		|27.3 M	|64.9	|87.0|
> |w/ LSA		|32.1		|23.8 M	|63.8	|86.4|
> |w/ Involution		|32.1		|23.8 M	|65.8	|87.5|
> |w/ Intervolution	|33.2		|24.1 M	|**66.3**	|**87.7**|
>
> Table C: Performance comparison with other spatio-temporal feature transform methods on Kinetics-400.
>
> Note that the results on Kinetics here are conducted in a simplified setting for comparison so that they do not reach the state of the art. The implementation details for our experiments are as follows. We adopt TSN-ResNet-50 as the backbone and replace one $3\times3$ conv layer with a single $5\times7\times7$ spatio-temporal feature transform layer. We adopt the dense frame sampling method [A6] and sample a clip of 8 frames. For training, we use a cosine learning rate schedule with the first 5 epochs for warm-up. We set the initial learning rate to 0.02 and total epochs to 50. For testing, we sample 10 uniform clips per video and average the softmax scores for the final prediction. We follow the strategy of non-local networks [A6] to pre-process the frames and take 3 crops as input.
>
> [A6] X. Wang *et al*., “Non-local Neural Networks,” CVPR, 2018.
>
> ***
>
>
> > [R iQAh] Backbone. The experiments in this paper mainly utilize ResNet50 as backbone. The results on more backbones could make the conclusions more convincing.
>
> Thanks for the constructive comments. As the reviewer suggested, we have validated the effectiveness of Intervolution on the 3D backbone, ResNet3D-18. We replace 8 $3\times3\times3$ convolutional layers in ResNet3D-18 by 8 $5\times7\times7$ Intervolution layers for every 2 layers. We set a learning rate as 0.1 and train models from scratch. The other training & testing details are the same as those in Sec.1 in the supplementary material. Table D shows the results that ResNet3D-18 with Intervolution shows 1.2\%p higher top-1 accuracy while significantly reducing FLOPs and the number of parameters by 10.3 G and 13.4 M, respectively.
>
> |model |FLOPs (G)|params (M)	|top-1 (%)	|top-5 (%)|
> |---|---|---|---|---|
> |ResNet3D-18		|41.3		|33.2		|40.8	|70.4|
> |w/ Intervolution	|31.0		|18.8		|**42.0**	|**71.6**|
>
> Table D: Performance of Intervolution with 3D ResNet-18 backbone on SS-v1.
>
> ***

---

> > ### Comment · Reviewer_iQAh · 2021-09-02
> > **Post-rebuttal feedback**
> >
> > Thanks for the detailed response and additional results. On one hand, the results on more datasets and more backbones seems to better validate the effectiveness of the proposed Intervolution. The result on Kinetics looks much lower than the other works. Considering the limited time, it is acceptable to me. On the other hand, I also agree that the writing should be improved and the idea is kind of incremental, which are raised by the other reviewers. Hence, I will keep my rating.

---

### Official Review · Reviewer_KG6i · 2021-07-16

**Rating:** 7
**Confidence:** 3

**Summary:**

The paper introduces a module that can process the spatio-temporal relations videos. The authors formalise the method as predicting for each point (target) in a feature map the output feature vector by aggregating a neighbourhood, acting as a context (of size $M \times C, M = T \times H \times W$) according to a dynamic kernel (of size M). Many models fit this framework, including convolution, self-attention, but the proposed method is shown experimentally to be superior. The paper introduces two main components: a *relational kernel* for aggregating the context, and *relational context* for improving the context, and shows that both help the performance of the model.


**Limitations And Societal Impact:**

The method shares the same Societal impact as any image/video classification method and it is addressed.

**Main Review:**

**Strong points**:

* The paper clearly presents the relation to prior work. The method is sound and the technical aspects are clearly explained.

* The method achieves superior performance on multiple datasets. The datasets are suitable for evaluating video processing methods and require modeling complex movements (e.g. for Smt-smt the motion is essential, the class cannot be predicted only from the general context)


* The paper presents good ablation studies, showing the importance of the proposed modules. The fair comparisons with other methods from Table 2 (especially local attention) are also appreciated.


**Weak Points**:

* Since omitting the softmax in the kernel seems to improve the performance, it will be useful to test LSA with the full kernel but without softmax (in table 2, just the variant with only content-to-position kernel is compared).


**Time Spent Reviewing:**

6

---

> ### Author Response · Authors · 2021-08-10
> **Response to Reviewer KG6i**
>
> Thanks for your constructive comments.
>
> ***
>
> > [R KG6i] Since omitting the softmax in the kernel seems to improve the performance, it will be useful to test LSA with the full kernel but without softmax (in table 2, just the variant with only content-to-position kernel is compared).
>
> As the reviewer suggested, we have evaluated LSA without softmax; instead of using softmax, we normalize $\boldsymbol{x}^\mathrm{Q}_n(\boldsymbol{X}^\mathrm{K}_n)^\top$ by dividing it by the size of local context $M=5\times 7\times 7$, following the dot-product instantiation of the non-local operation [A1]. The results are summarized below. The LSA without softmax nonlinearity achieves 42.4 \% at top-1 accuracy, which is slightly higher as 0.8 \%p than the original LSA but lower than $\boldsymbol{x}^\mathrm{Q}_n\boldsymbol{P}\boldsymbol{X}^\mathrm{V}_n$. We conjecture that an appropriate normalization technique should be applied to $\boldsymbol{X}^\mathrm{K}_n$ (Table 12 in [A2]) to improve the performance. We think that it needs to be more studied.
>
> |model			|kernel		|context	 |FLOPs (G)	|Params (M)	|top-1 (%)	|top-5 (%)|
> |--|--|--|--|--|--|--|
> |w/ LSA			|$\sigma(\boldsymbol{x}_n^{\mathrm{Q}}(\boldsymbol{X}_n^{\mathrm{K}})^\top + \boldsymbol{x}_n^{\mathrm{Q}} \boldsymbol{P}^\top)$	|$\boldsymbol{X}_n^{\mathrm{V}}$	|32.2		|24.3		|41.6	|70.9|
> |w/ LSA variant			|$\sigma(\boldsymbol{x}_n^{\mathrm{Q}} \boldsymbol{P}^\top)$	|$\boldsymbol{X}_n^{\mathrm{V}}$	|32.1		|23.4		|41.3	|70.6|
> |w/ LSA	variant		|$\boldsymbol{x}_n^{\mathrm{Q}}(\boldsymbol{X}_n^{\mathrm{K}})^\top + \boldsymbol{x}_n^{\mathrm{Q}} \boldsymbol{P}^\top$		|$\boldsymbol{X}_n^{\mathrm{V}}$	|32.2		|23.4		|42.4	|72.1|
> |w/ LSA	variant		|$\boldsymbol{x}_n^{\mathrm{Q}} \boldsymbol{P}^\top$		|$\boldsymbol{X}_n^{\mathrm{V}}$	|32.1		|23.4		|44.3	|73.7|
> |w/ intervolution	|$\boldsymbol{\kappa}_n^{\mathrm{V}} + \boldsymbol{\kappa}_n^{\mathrm{R}}$	|$\boldsymbol{X}_n^{\mathrm{V}}$	|32.6		|23.4		|45.7	|74.8|
> |w/ intervolution	|$\boldsymbol{\kappa}_n^{\mathrm{V}} + \boldsymbol{\kappa}_n^{\mathrm{R}}$	|$\boldsymbol{X}_n^{\mathrm{V}} + \boldsymbol{X}_n^{\mathrm{R}}$	|33.2		|23.6		|**47.0**|**75.7**|
>
>
> [A1] X. Wang *et al*., “Non-local Neural Networks,” CVPR, 2018.\
> [A2] I. Bello, “LambdaNetworks: Modeling Long-range Interactions without Attention,” ICLR, 2021.

---

> > ### Comment · Reviewer_KG6i · 2021-09-02
> > **Response to rebuttal**
> >
> > I thank the authors for their clarifications and extra experiments.
> >
> > After reading the other reviews, I agree that the paper could be more clear and terms like content-to-content or content-to-position should be better explained. At the moment they are mainly self-explained by the terms in the equations but this could be addressed to some degree in the final version.
> >
> > As the paper offers a method for better handling of interactions in videos, I think that leaving out Kinetics is ok, as Kinetics relies more heavily on appearance as opposed to motion and long-range interactions that are better captured in datasets like Smt-Smt.  The results on Smt-Smt are SOTA, so I cannot say they are not significant. Also, the authors provided good ablations in the main paper and in the rebuttal.
> >
> > I think the paper has value in exploring attention architectures and making connections and good comparisons to prior attention-based / Transformer based models. I will keep my initial rating.

---

### Official Review · Reviewer_rPEM · 2021-07-18

**Rating:** 5
**Confidence:** 4

**Summary:**

The authors propose a novel approach for action recognition. They integrate the ideas from current popular operators: convolution, involution, and self-attention into a relational feature transform. The new method can capture the spatial-temporal relations for video understanding, and it achieves state-of-the-art results on the motion-centric action recognition benchmark.

**Ethics Review Area:**

["I don’t know"]

**Limitations And Societal Impact:**

1. I can tell the authors try very hard to describe the motivation of intervolution from convolution, self-attention, and involution. But it still takes me some time to figure out the idea of the relational kernel, relational context, and some other concepts like content-to-content and content-to-position. I am wondering whether the author could simplify the presentation of the method so that the readers can easily interpret and reproduce the method.
2. This paper is targeted to improve the relational representation; thus, the experiments are mostly applied on motion-centric datasets. However, the readers would like to see the comparison results on the other action recognition datasets like Kinetics.
3. I know there is a huge literature working on this topic. However, please cover some representative methods: TRN [1], TPN [2] as well as the recent ones: MoviNet [3], etc.

[1] Zhou, Bolei, et al. "Temporal relational reasoning in videos." Proceedings of the European Conference on Computer Vision (ECCV). 2018.
[2] Yang, Ceyuan, et al. "Temporal pyramid network for action recognition." Proceedings of the IEEE/CVF Conference on Computer Vision and Pattern Recognition. 2020.
[3] Kondratyuk, Dan, et al. "Movinets: Mobile video networks for efficient video recognition." Proceedings of the IEEE/CVF Conference on Computer Vision and Pattern Recognition. 2021.

**Main Review:**

1. The authors address the problems in current popular operators for learning video representations. The motivation is clear and the idea is interesting.
2. The method is lightweight and effective. It can reduce the computation cost and improve the accuracy of action recognition at the same time.
3. The proposed approach can achieve SOTA results on public datasets like SS-V1&V2 with very light computation costs (GFLOPs). The experiments on motion-centric benchmarks and the ablation study are sufficient.


**Time Spent Reviewing:**

6 hours

---

> ### Author Response · Authors · 2021-08-10
> **Response to Reviewer rPEM**
>
> Thanks for your constructive comments.
>
> ***
>
> > [R rPEM] I can tell the authors try very hard to describe the motivation of intervolution from convolution, self-attention, and involution. But it still takes me some time to figure out the idea of the relational kernel, relational context, and some other concepts like content-to-content and content-to-position. I am wondering whether the author could simplify the presentation of the method so that the readers can easily interpret and reproduce the method.
>
> Thanks for the suggestion. We will do our best to revise the explanation and the figures for the final version by reflecting the reviewers’ comments.
>
> ***
>
> > [R rPEM] This paper is targeted to improve the relational representation; thus, the experiments are mostly applied on motion-centric datasets. However, the readers would like to see the comparison results on the other action recognition datasets like Kinetics.
>
> Following the reviewer’s suggestion, we conducted an additional experiment on Kinetics-400 dataset, which is one of the appearance-centric action benchmarks. Table A below summarizes the results, which show that Intervolution outperforms other feature transform methods when the same size of the local context is used. We validate that the relational kernel and the relational context still show a clear gain on the appearance-centric benchmark.
>
> |model (TSN-R50)	|FLOPs (G)		|Params (M)	|top-1 (%)	|top-5 (%)|
> |--|--|--|--|--|
> |w/ R(2+1)D		|37.3		|27.3 M		|64.9	|87.0|
> |w/ LSA		|32.1		|23.8 M		|63.8	|86.4|
> |w/ Involution		|32.1		|23.8 M		|65.8	|87.5|
> |w/ Intervolution	|33.2		|24.1 M		|**66.3**	|**87.7**|
>
> Table A: Performance comparison with other spatio-temporal feature transform methods on Kinetics-400.
>
> Note that the results on Kinetics here are conducted in a simplified setting for comparison so that they do not reach the state of the art. The implementation details for our experiments are as follows. We adopt TSN-ResNet-50 as the backbone and replace one $3\times 3$ conv layer with a single $5\times 7\times 7$ spatio-temporal feature transform layer. We adopt the dense frame sampling method [A3] and sample a clip of 8 frames. For training, we use a cosine learning rate schedule with the first 5 epochs for warm-up. We set the initial learning rate to 0.02 and total epochs to 50. For testing, we sample 10 uniform clips per video and average the softmax scores for the final prediction. We follow the strategy of non-local networks [A3] to pre-process the frames and take 3 crops as input.
>
> [A1] X. Wang et al., “Non-local Neural Networks,” CVPR, 2018.
>
> ***
>
> > [R rPEM] I know there is a huge literature working on this topic. However, please cover some representative methods: TRN [1], TPN [2] as well as the recent ones: MoviNet [3], etc.
>
> Thanks for the references. We will discuss them in Section 2 of the manuscript.

---

> > ### Comment · Reviewer_rPEM · 2021-08-29
> > **Doubt of the novelty and contributions.**
> >
> > I appreciate the additional comparison results and the authors addressed some of my questions. Actually I am surprised when I saw the positive feedbacks from the other reviewers and I am the only one who give the negative score to this paper. Thus I read the paper again. I still can not raise my score since i do not think the quality of the paper meet the standard of NeurIPS. The reasons can be listed as follows:
> > 1. The writing is not good and the organization of the paper is not clear. For example, to the best of the my knowledge, I did not see the phrase like 'content-to-content' and 'content-to-position' in the literature of action recognition and the authors did not make a good explanation for those terminologies.
> > 2. I like the results on Something-Something. But the improvement is not that significant. The authors provide the comparison results on Kinetics during rebuttal period. However, the results are not solid since they only compare with weak baselines.
> > 3. The idea of intervolution is kind of incremental.
> > 4. I am still hesitating to give my final score and I may change my mind if the authors and the other reviewers can persuade me.

---

### Official Review · Reviewer_1xyW · 2021-07-19

**Rating:** 6
**Confidence:** 4

**Summary:**

The paper proposes a dynamic feature transform method named intervolution. While bearing similarities with involution and lambda convolution, the proposed intervolution proposes to add content-to-content relational learning in both dynamic kernel generation and feature value transforms. Alongside the basic kernel and context transforms, additional learnable parameters are introduced for the relational kernel and context learning. To further improve the efficiency of intervolution, decomposition of the projection matrix and the use of multiple queries are proposed. The experiments use TSN-ResNet50 baseline and replace the 3D convolution layers with intervolution layers. Experiments on SS-v1 and v2, Diving-48 and FineGym datasets show that intervolution provides a reasonable efficiency-accuracy trade-off.

**Limitations And Societal Impact:**

The authors have adequately addressed the limitations and potential negative societal impact of their work.

**Main Review:**

* The proposed relational kernel and context learning is effective in learning spatial-temporal interactions compared to the static as well as existing dynamic feature transform methods.

* Intervolution introduces not much of extra FLOPs, number of parameters while achieving consistent improvements on different action recognition benchmarks.

* What does memory usage look like in Table 2?

* It seems like the current method exploits CUDA implementation. Does it affect FLOPs or memory usage without CUDA modification? (e.g. pure Pytorch style implementation).


**Time Spent Reviewing:**

4 hours

---

> ### Author Response · Authors · 2021-08-10
> **Response to Reviewer 1xyW**
>
> Thanks for your constructive comments.
>
> ***
>
> > [R 1xyW] What does memory usage look like in Table 2?
>
> In the table below, we show the memory footprint of the models. Note that LSA and its variants consume a large amount of memory for generating attention maps, while our Intervolution methods are memory-efficient since ours do not generate attention maps by switching permutation orders (L37-49 in Supp.).  Please refer to pseudo-codes in Fig.2 in our supplementary material 1 and their actual PyTorch implementations (Intervolution_NeurIPS2021/ops/intervolution.py) posted in the zip file. Furthermore, since we find out the FLOPs typo (32.6 G -> 33.2 G) in the last row of Table 2, we modify it in the table below. We will fix it in the final manuscript.
>
> |model 			|kernel		|context		|FLOPs (G)	 	|Memory (GB)	|top-1 (%)|
> |--|--|--|--|--|--|
> |C2D			|$\boldsymbol{W}_{2D}$		|$\boldsymbol{X}_n^{\mathrm{V}}$		|32.5		|7.4		|19.7|
> |w/ C3D		|$\boldsymbol{W}_{3D}$		|$\boldsymbol{X}_n^{\mathrm{V}}$		|57.0		|10.3	|43.4|
> |w/ R(2+1)D		|$\boldsymbol{W}_{(2+1)D}$	|$\boldsymbol{X}_n^{\mathrm{V}}$		|37.3		|7.5		|44.1|
> |w/ LSA		|$\sigma(\boldsymbol{x}_n^{\mathrm{Q}}(\boldsymbol{X}_n^{\mathrm{K}})^\top + \boldsymbol{x}_n^{\mathrm{Q}} \boldsymbol{P}^\top)$	|$\boldsymbol{X}_n^{\mathrm{V}}$		|32.2		|9.0		|41.6|
> |w/ LSA	variant		|$\sigma(\boldsymbol{x}_n^{\mathrm{Q}} \boldsymbol{P}^\top)$	|$\boldsymbol{X}_n^{\mathrm{V}}$		|32.1		|8.7		|41.3|
> |w/ LSA	variant		|$\boldsymbol{x}_n^{\mathrm{Q}} \boldsymbol{P}^\top$		|$\boldsymbol{X}_n^{\mathrm{V}}$		|32.1		|8.6		|44.3|
> |w/ intervolution	|$\boldsymbol{\kappa}_n^{\mathrm{V}} + \boldsymbol{\kappa}_n^{\mathrm{R}}$	|$\boldsymbol{X}_n^{\mathrm{V}}$		|32.6		|7.6		|45.7|
> |w/ intervolution	|$\boldsymbol{\kappa}_n^{\mathrm{V}} + \boldsymbol{\kappa}_n^{\mathrm{R}}$	|$\boldsymbol{X}_n^{\mathrm{V}} + \boldsymbol{X}_n^{\mathrm{R}}$		|33.2		|7.8		|**47.0**|
>
>
> Table A: Performance comparison with other spatio-temporal feature transform methods on SS-v1.
>
> ***
>
> > [R 1xyW] It seems like the current method exploits CUDA implementation. Does it affect FLOPs or memory usage without CUDA modification? (e.g. pure Pytorch style implementation)
>
> We implement Intervolution without any CUDA implementation; we make it efficient by permuting the computation order of Intervolution from eq.3 to eq.4 in our supplementary material (L37-49 in Supp.). Please refer to pseudo-codes in Fig.2 in supplementary material and their actual PyTorch implementation (Intervolution_NeurIPS2021/ops/intervolution.py) posted in the zip file.

---

> > ### Comment · Reviewer_1xyW · 2021-09-01
> > **Post-rebuttal feedback**
> >
> > I have read other reviews and the authors rebuttal. I thank the reviewers' reply to the question on the actual memory usage and implementation of the intervolution. I keep my original rating.

---

### Decision · Program_Chairs · 2021-09-28

**Decision:**

Accept (Poster)

**Comment:**

The paper received three borderlines (1 negative, 2 positive) and one accept ratings. The main weaknesses were:
1. The results on Kinetics are weak.
2.  The ideas are kind of incremental.
3.  The presentation of this paper is unclear.

* The AC does not consider 1 an important concern. It would be nice to have strong results on Kinetics as well, but kinetics relies more on appearance rather than motion and long-range interactions  (the focus of this paper).
* The AC shares the same concern as 2 and 3. The technical contributions and the writing quality do not meet the bar of NeurIPS.

Due to these issues, the AC recommends rejection. The authors are encouraged to consider the reviewers' comments when revising the paper for submission elsewhere.

**Consistency Experiment:**

NeurIPS has a long history of experimentation. In 2014, NeurIPS ran an experiment in which 10% of submissions were reviewed by two independent committees to quantify the randomness in the review process. This year, we repeated a variant of this experiment to see how the quality of the review process has changed over time.  This paper was part of the experiment and was therefore assigned to two committees (consisting of reviewers, an Area Chair, and a Senior Area Chair) that reached independent decisions.  If both committees made the same recommendation, this recommendation was followed. If a single committee recommended acceptance, the paper was accepted (with the exception of a few cases in which the other committee identified what we considered a fatal flaw, e.g., an error in a key result).

This copy’s committee reached the following decision: **Reject**

The other committee assigned to the paper recommended **Accept (Spotlight)**.  You can find the other set of reviews, along with any follow up discussion with the authors here:
https://openreview.net/forum?id=nwu1RUCkei4